# Preclinical characterization of an intravenous coronavirus 3CL protease inhibitor for the potential treatment of COVID19

Britton Boras[1,17], Rhys M. Jones [1,17✉], Brandon J. Anson [2], Dan Arenson[3], Lisa Aschenbrenner[3], Malina A. Bakowski [4], Nathan Beutler [5], Joseph Binder[1], Emily Chen[4], Heather Eng [3], Holly Hammond[6], Jennifer Hammond[7], Robert E. Haupt[6], Robert Hoffman [1], Eugene P. Kadar[3], Rob Kania[1], Emi Kimoto [3], Melanie G. Kirkpatrick[4], Lorraine Lanyon[3], Emma K. Lendy[8], Jonathan R. Lillis[9], James Logue [6], Suman A. Luthra[10], Chunlong Ma [11], Stephen W. Mason[12,3], Marisa E. McGrath [6], Stephen Noell[3], R. Scott Obach[3], Matthew N. O' Brien[13], Rebecca O'Connor[3], Kevin Ogilvie [3], Dafydd Owen[10], Martin Pettersson[10], Matthew R. Reese [3], Thomas F. Rogers [5,14], Romel Rosales [15,16], Michelle I. Rossulek[10], Jean G. Sathish[12], Norimitsu Shirai[3], Claire Steppan[3], Martyn Ticehurst[9], Lawrence W. Updyke[10], Stuart Weston [6], Yuao Zhu[12], Kris M. White [15,16], Adolfo García-Sastre [15,16], Jun Wang [11], Arnab K. Chatterjee[4], Andrew D. Mesecar[2,8], Matthew B. Frieman [6], Annaliesa S. Anderson [12] & Charlotte Allerton [10]

COVID-19 caused by the SARS-CoV-2 virus has become a global pandemic. 3CL protease is a virally encoded protein that is essential across a broad spectrum of coronaviruses with no close human analogs. PF-00835231, a 3CL protease inhibitor, has exhibited potent in vitro antiviral activity against SARS-CoV-2 as a single agent. Here we report, the design and characterization of a phosphate prodrug PF-07304814 to enable the delivery and projected sustained systemic exposure in human of PF-00835231 to inhibit coronavirus family 3CL protease activity with selectivity over human host protease targets. Furthermore, we show that PF-00835231 has additive/synergistic activity in combination with remdesivir. We present the ADME, safety, in vitro, and in vivo antiviral activity data that supports the clinical evaluation of PF-07304814 as a potential COVID-19 treatment.

[1] Worldwide Research and Development, Pfizer Inc, La Jolla, CA 92121, USA. [2] Department of Biological Sciences, Purdue University, West Lafayette, IN 47907, USA. [3] Worldwide Research and Development, Pfizer Inc, Groton, CT 06340, USA. [4] Calibr, a division of The Scripps Research Institute, La Jolla, CA 92037, USA. [5] Department of Immunology and Microbiology, The Scripps Research Institute, La Jolla, CA 92037, USA. [6] Department of Microbiology and Immunology University of Maryland School of Medicine, Baltimore, MD 21201, USA. [7] Worldwide Research and Development, Pfizer Inc., Collegeville, PA 19426, USA. [8] Department of Biochemistry, Purdue University, West Lafayette, IN 47907, USA. [9] Worldwide Research and Development, Pfizer Inc, Sandwich CT13 9ND, UK. [10] Worldwide Research and Development, Pfizer Inc, Cambridge, MA 02139, USA. [11] Department of Pharmacology and Toxicology, College of Pharmacy, University of Arizona, Tucson, AZ 85721, USA. [12] Worldwide Research and Development, Pfizer Inc., Pearl River, NY 10965, USA. [13] Worldwide Research and Development, Pfizer Inc, Lake Forest, IL 60045, USA. [14] UC San Diego Division of Infectious Diseases and Global Public Health, UC San Diego School of Medicine, La Jolla, CA 92093, USA. [15] Department of Microbiology, Icahn School of Medicine at Mount Sinai, New York, NY, USA. [16] Global Health and Emerging Pathogens Institute, Icahn School of Medicine at Mount Sinai, New York, NY, USA. [17]These authors contributed equally: Britton Boras, Rhys M. Jones. ✉email: rhys.jones@pfizer.com

In December 2019, COVID-19 was identified as a new, potentially fatal, respiratory infection caused by severe acute respiratory syndrome coronavirus 2 (SARS-CoV-2)[1,2]. Unlike previous coronavirus outbreaks that receded relatively quickly, the resultant COVID-19 pandemic spread across the globe. To date, over 100 million people have been infected and millions of people have died globally.

The RNA-dependent RNA polymerase (RdRp) inhibitor remdesivir is currently the only clinically approved drug for the treatment of SARS-CoV-2 and was granted emergency use authorization by the U.S. Food and Drug Administration (FDA) in May 2020[3]. To date, the results of trials of remdesivir in hospitalized patients have been mixed[4] despite being granted full authorization in October 2020[5]. Monoclonal antibodies that target the viral spike protein have also been successful in treating disease when applied at the early stage of infection[6]. Though the availability of highly efficacious vaccines provides great hope for the eradication of COVID-19, it is expected that it will still take time for cases to subside globally and does not address the potential threat of future coronaviruses. Thus, other classes of antivirals that exhibit single-agent efficacy or that are complementary to remdesivir for use in combination regimens are essential to meet this substantial unmet need.

SARS-CoV-2 produces two large viral polyproteins, pp1a and pp1ab, which are processed by two virally encoded cysteine proteases, the main protease, also called 3C-like protease (3CL protease or 3CL^pro) and the papain-like protease. Mutagenesis experiments with other coronaviruses have demonstrated that the activity of the 3CL^pro is essential for viral replication[7,8]. 3CL^pro proteolytically processes the virus p1a/p1ab polyproteins at more than 10 junctions to generate a series of non-structural proteins critical for virus replication and transcription, including RdRp, the helicase, and the 3CL^pro itself[9]. No close human analogs of the coronavirus 3CL^pro are known, suggesting that selective 3CL^pro inhibitors should avoid unwanted polypharmacology[10]. The essential functional importance of proteases in virus replication has led to the clinical success of protease inhibitors in combating both human immunodeficiency virus (HIV) and hepatitis C virus (HCV)[11–13]. This together with the opportunity for selectivity, makes 3CL^pro an attractive antiviral drug target[14].

Following the severe acute respiratory syndrome (SARS) outbreak in 2002–2003 we identified a potential small molecule protease inhibitor (PF-00835231) for the treatment of SARS-CoV, using structure-based drug design[15]. Due to the SARS pandemic being brought under control in July 2003 following public health measures which incorporated patient isolation and travel restrictions, this project was discontinued due to the lack of a path forward to demonstrate clinical efficacy. Given that the SARS-CoV and SARS-CoV-2 3CL^pro sequences share 96%

identity overall and 100% identity in the active site[1,16,17], following the emergence of SARS-CoV-2, PF-00835231 was identified as a potential SARS-CoV-2 3CL^pro inhibitor for the treatment of COVID-19 disease[15,18]. Subsequent antiviral data presented here indicate that PF-00835231 has similar potency against either SARS-CoV or SARS-CoV-2 (Table S1).

Herein we describe the 3CL^pro inhibitor, PF-00835231, and the preclinical characterization of its phosphate prodrug, PF-07304814, and present broad-spectrum inhibitory activity across coronaviral 3CL proteases, in vitro and in vivo antiviral activity in a coronavirus animal model, and absorption, distribution, metabolism, excretion (ADME) and safety data highlighting its potential for the intravenous (IV) treatment of COVID-19 disease.

## Results

**PF-00835231 exhibits tight and specific binding to SARS-CoV-2 3CL in vitro.** A thermal-shift assay was used to evaluate the direct binding between PF-00835231 and SARS-CoV-2 3CL^pro. The melting temperature of SARS-CoV-2 3CL^pro was shifted by 14.6 °C upon binding of PF-00835231, from $55.9 \pm 0.11$ °C ($n = 16$) to $70.5 \pm 0.12$ °C ($n = 8$). These data support tight and specific binding of PF-00835231 to SARS-CoV-2 3CL^pro (Fig. 1) as was shown previously by X-ray co-crystal structure and evaluation of $K_i$[15].

**PF-00835231 has potent and broad-spectrum inhibitory activity against a panel of coronavirus 3CL^pros.** To explore the notion that PF-00835231 could have pan-coronavirus activity, PF-00835231 was evaluated against 3CL^pro from a variety of other coronaviruses representing alpha, beta and gamma groups of *Coronaviridae*, using biochemical Förster Resonance Energy Transfer (FRET) protease activity assays. PF-00835231 demonstrated potent inhibitory activity against all tested coronavirus 3CL^pro including members of alpha-coronaviruses (NL63-CoV, HCoV-229E, PEDV, FIPV), beta-coronaviruses (HKU4-CoV, HKU5-CoV, HKU9-CoV, MHV-CoV, OC43-CoV, HKU1-CoV), and gamma-coronavirus (IBV-CoV), with $K_i$ values ranging from 30 pM to 4 nM (Table 1). The demonstrated activity is consistent with a potential therapeutic use against emerging coronaviruses. This inhibitory activity is restricted to coronavirus 3CL^pros as PF-00835231 was inactive against a panel of human proteases and HIV protease (Table S2). PF-00835231 showed detectable activity against human cathepsin B but 1000-fold

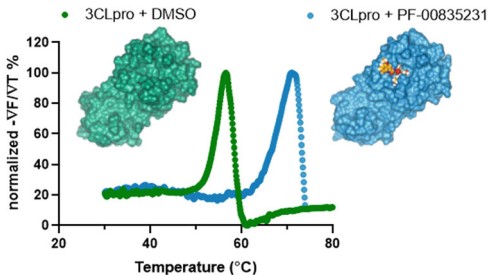

**Fig. 1 Representative thermal shift binding data of PF-00835231 with SARS-CoV-2 3CL^pro.** X-ray structures of SARS CoV-2 3CL^pro apoenzyme (left) and SARS CoV-2 3CL^pro in complex with PF-00835231 (right). Representative of melting curves with and without PF-00835231 ($n = 8$, $n = 16$ respectively).

**Table 1 Activity of PF-00835231 against 3CL^pro of coronaviruses (mean ± SEM).**

| Virus | $K_i$ (nM) |
|---|---|
| *Alpha-CoV* | |
| NL63-CoV | 0.77 ± 0.52 |
| HCoV-229E | 1.5 ± 0.76 |
| PEDV | 0.30 ± 0.11 |
| FIPV | 0.12 ± 0.10 |
| *Beta-CoV* | |
| SARS-CoV-2^a | 0.27 ± 0.1 |
| HKU1-CoV | 0.85 ± 0.24 |
| HKU4-CoV | 0.034 ± 0.079 |
| HKU5-CoV | 0.033 ± 0.12 |
| HKU9-CoV | 0.74 ± 0.68 |
| MHV-CoV | 1.2 ± 0.90 |
| OC43-CoV | 0.51 ± 0.12 |
| *Gamma-CoV* | |
| IBV-CoV | 4.0 ± 0.37 |

^aData reported in ref. [15].

**A**

| Cells | Virus | Efflux Inhibitor | PF-00835231 | | PF-07304814 | |
|---|---|---|---|---|---|---|
| | | | EC$_{50}$ μM GeoMean (95% CI) | CC$_{50}$ μM GeoMean (95% CI) | EC$_{50}$ μM GeoMean (95% CI) | CC$_{50}$ μM GeoMean (95% CI) |
| Vero E6-enACE2 (kidney) | SARS2 Washington 1 | 0 | 39.8 (29.8,53.2) n=10 | >100 (ND) n=5 | 86.7 (71,106) n=10 | >100 (ND) n=4 |
| | | 0.5 μM | 2.93 (1.13,7.64) n=7 | >100 (ND) n=2 | 26.6 (7.6,93.6) n=7 | >100 (ND) n=3 |
| | | 2 μM | 0.236 (0.135,0.412) n=6 | >100 (ND) n=2 | 3.8 (1.6,8.8) n=7 | >100 (ND) n=2 |
| VeroE6-EGFP (kidney) | SARS2 BetaCov GHB-03021/2020 | 0 | 88.9 (76.8,103) n=5 | >100 (ND) n=3 | >50 (ND) n=2 | >50 (ND) n=2 |
| | | 0.5 μM | >8.21 (3.12, 21.6) n=8 | >100 (ND) n=3 | 27 (6.3, 116) n=2 | >50 (ND) n=2 |
| | | 2 μM | 0.76 (0.45,1.29) n=4 | >50 (ND) n=2 | 0.83 (0.50,1.37) n=3 | >50 (ND) n=2 |
| MRC-5 (lung) | HCoV-229E | 0 | 0.069 (0.056,0.085) n=7 | >100 (ND) n=2 | 0.074 (0.013, 0.417) n=3 | >100 (ND) n=3 |
| | | 0.5 μM | 0.080 (0.017, 0.37) n=3 | >100 (ND) n=2 | 0.058 (0.023, 0.15) n=3 | >100 (ND) n=3 |

**B**

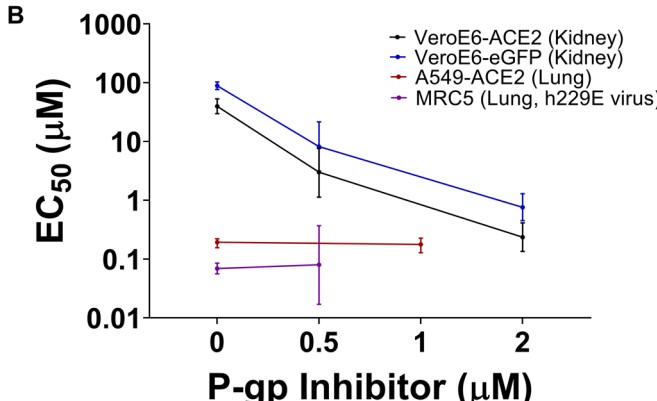

**Fig. 2 Antiviral activity across cell lines and viruses. A** In vitro antiviral activity (EC$_{50}$), and cytotoxicity (CC$_{50}$) for PF-00835231 and PF-07304814 with and without the P-gp efflux inhibitor, CP-100356. *n* as shown reflects individual replicates (**B**) EC$_{50}$ values with PF-00835231 with increasing P-gp inhibitor in human lung and monkey kidney cell lines. A549-ACE2 human lung carcinoma data (red) as reported in ref. [18]. Data are presented as mean±95% confidence interval as error bars.

weaker (6.9 nM vs 6 μM) activity compared to 3CL$^{pro}$ (Table S2). Thereby, these data collectively support PF-00835231 as a selective in vitro protease inhibitor with broad coronavirus activity.

**In vitro cellular antiviral activity of PF-00835231 against SARS-CoV-2.** The antiviral activity of PF-00835231 against SARS-CoV-2 in cell culture was evaluated with a cytopathic effect (CPE) assay using either VeroE6 kidney cells enriched for angiotensin-converting enzyme 2 (ACE2) (VeroE6-enACE2) receptor or VeroE6 cells constitutively expressing EGFP (VeroE6-EGFP). These cell lines were infected with the SARS-CoV-2 Washington strain 1 (WA1-EPI_ISL_404895) or the Belgium/GHB-03021/2020 strain (GHB-03021-EPI_ISL_407976)[19], respectively, which have identical 3CL$^{pro}$ amino acid sequences. PF-00835231 exhibited viral CPE EC$_{50}$ values of 39.7 μM and 88.9 μM, respectively (EC$_{50}$, Fig. 2). However, Vero cells express high levels of the efflux transporter P-glycoprotein (P-gp) (also known as MDR1 or ABCB1), of which PF-00835231 is a known substrate[15] suggesting that the intracellular concentration of PF-00835231 was lower than it initially appeared. Therefore, to evaluate the full potency of PF-00835231, the assays were repeated in the presence of a P-gp efflux inhibitor, CP-100356[20]. PF-00835231 exhibited a 117- to 173-fold increase in activity in the presence of 2 μM P-gp inhibitor, with EC$_{50}$ values of 0.23 μM in VeroE6-enACE2 cells and 0.76 μM in the VeroE6-EGFP cells (Fig. 2). The P-gp inhibitor alone had no antiviral or cytotoxic activity at these concentrations and did not cause cytotoxicity in the presence of the protease inhibitor (Table S3). The use of

VeroE6 cells by many in the field to evaluate inhibitor activity could be problematic, since the true activity of some compounds could be masked by efflux of inhibitors from these cells (Fig. 2B). Consistent with many viral protease inhibitors[21], there was a steep response to increasing doses of PF-00835231, with a ~2–3 fold difference between EC$_{50}$ and EC$_{90}$ in both cell types (EC$_{90}$ = 0.48 μM in VeroE6-enACE2 cells and EC$_{90}$ = 1.6 μM in VeroE6-EGFP cells) in the presence of the P-gp inhibitor. As expected, when lung cell lines were tested for antiviral potency in the presence and absence of P-gp inhibitor (A549-ACE2[18] and MRC5), no statistical difference in antiviral potency was observed (Fig. 2A). Additionally, antiviral activities in both VeroE6 cell lines with 2 μM P-gp inhibitor are similar to those observed in more physiologically relevant human lung cell culture systems, including A549-ACE2 and polarized human airway epithelial cells[18], where P-gp expression is lower. These data support the potential for single agent antiviral activity. As will be presented later, despite seeing apparent antiviral activity for PF-07304814, a prodrug of PF-00835231, in cell-based assays (Fig. 2A) this activity is likely due to the conversion of PF-07304814 to PF-00835231 in the assay.

**Potential for antiviral combination benefit of PF-00835231 in combination with remdesivir.** Combinations of antiviral agents, especially those targeting different steps in the virus replication cycle, are a frequently employed therapeutic strategy in treating viral diseases[22]. As PF-00835231 and remdesivir, a nucleoside RNA-dependent RNA polymerase inhibitor, target different steps

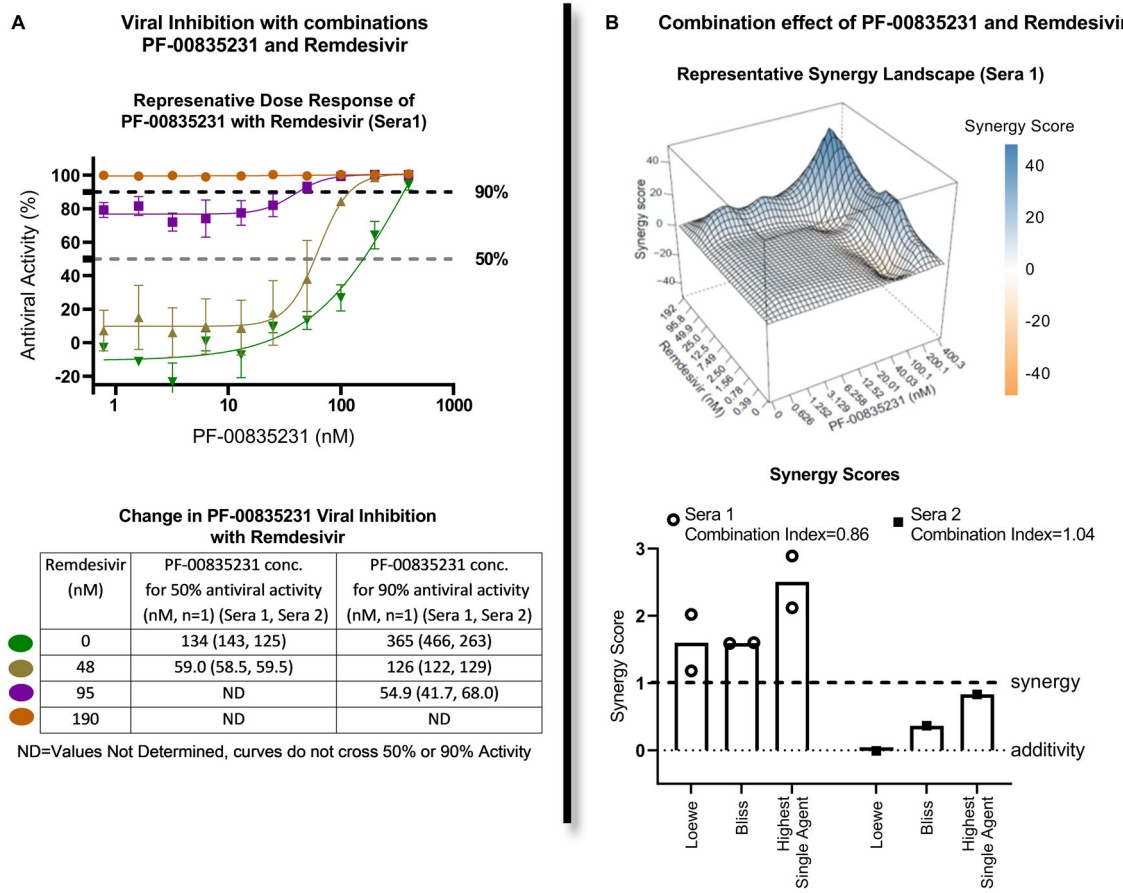

**Fig. 3 Measuring potential synergy between PF-00835231 and remdesivir in HeLa-ACE2 cells. A** (Top) Representative antiviral dose–response curves of PF-00835231 in combination with remdesivir against SARS-CoV-2. Serial dilutions of PF-00835231 with a range of fixed concentrations of remdesivir. Mean±SD from three technical replicates. Representative of three independent experiments. (Bottom) In vitro average absolute antiviral activity shift in 50% and 90% antiviral activity with fixed concentrations of remdesivir. Average from three technical replicates each of two different patient convalescent sera. Representative of three independent experiments. **B** (Top) A representative three-dimensional drug interaction landscape plotting Zero Interaction Potency synergy scores analyzed using Synergyfinder (median scores of three replicates). (Bottom) Average in vitro combination synergy scores from the three experiments using two different patients' sera (shown separately).

in the virus replication cycle, the antiviral activity of the two compounds was evaluated alone and in combination using HeLa-ACE2 cells in a high-content imaging assay[23]. Viral proteins were detected in this assay using convalescent human polyclonal sera from two different COVID-19 patients followed by fluorescent secondary antibodies and imaging[24]. PF-00835231 alone inhibited SARS-CoV-2 replication with an average $EC_{50}$ of 0.13 μM and $EC_{90}$ of 0.43 μM, whereas remdesivir had an average $EC_{50}$ of 0.074 μM and $EC_{90}$ of 0.17 μM (Fig. 3A). Combination studies were performed using a drug testing matrix, and the data for the drug combination were analyzed using reference models (Loewe, Bliss, HSA) to classify the effects of the drug combination as either additive, synergistic or antagonistic (isobologram, synergy scores, and combination indices).

As summarized in Fig. 3B, the combination of PF-00835231 and remdesivir using patient sera for detection, exhibited synergy in two independent experiments with sera from patient #1 and additivity in a single experiment with sera from patient #2 (Fig. 3B). The different classification is most likely due to the different convalescent serum used as detection reagents. These same antiviral data were also analysed using Synergyfinder, which also indicated that the two drugs were additive to synergistic, with a representative graph shown in Fig. 3B. Antagonism was not demonstrated for the combination of PF-00835231 and remdesivir in these studies. The observed additivity/synergy was not due

to cytotoxicity, as there was no noticeable cytotoxicity in virus-infected host cells for all the combinations tested. This additivity/synergy is similar to other protease inhibitors used for the treatment of HCV and HIV, which has led to substantial clinical benefit[25].

**Activity of PF-00835231 in mouse models of SARS-CoV and SARS-CoV-2 infection.** Human coronaviruses can replicate in mice but do so poorly and do not cause disease due to the specificity of the viral spike fusion proteins and differences between the human and mouse orthologs of the primary SARS-CoV and CoV-2 receptor (ACE2). To circumvent this we used two different models, one with a mouse-adapted (MA) SARS-CoV strain[26] and a second by supplying the human ACE2 receptor in trans to facilitate SARS-CoV-2 infectivity[27]. Given the sequence identity between SARS-CoV and SARS-CoV-2 3CL proteases, especially in the active site (100% identical), it was expedient to evaluate PF-00835231 in the validated SARS-CoV MA15 model[19]. SARS-CoV-MA15 has two mutations in the 3CL protease coding sequence[28] that are located distal to the active site that did not influence activity of the protease or its inhibition by PF-00835231 (Table S1). Treatment of MA15-infected BALB/c mice with 100 mg/kg PF-00835231 twice daily (BID) by subcutaneous (S.C.) injection, started either at the same time as infection, i.e., day 0, or on day 1 or 2 post-infection. Lung viral

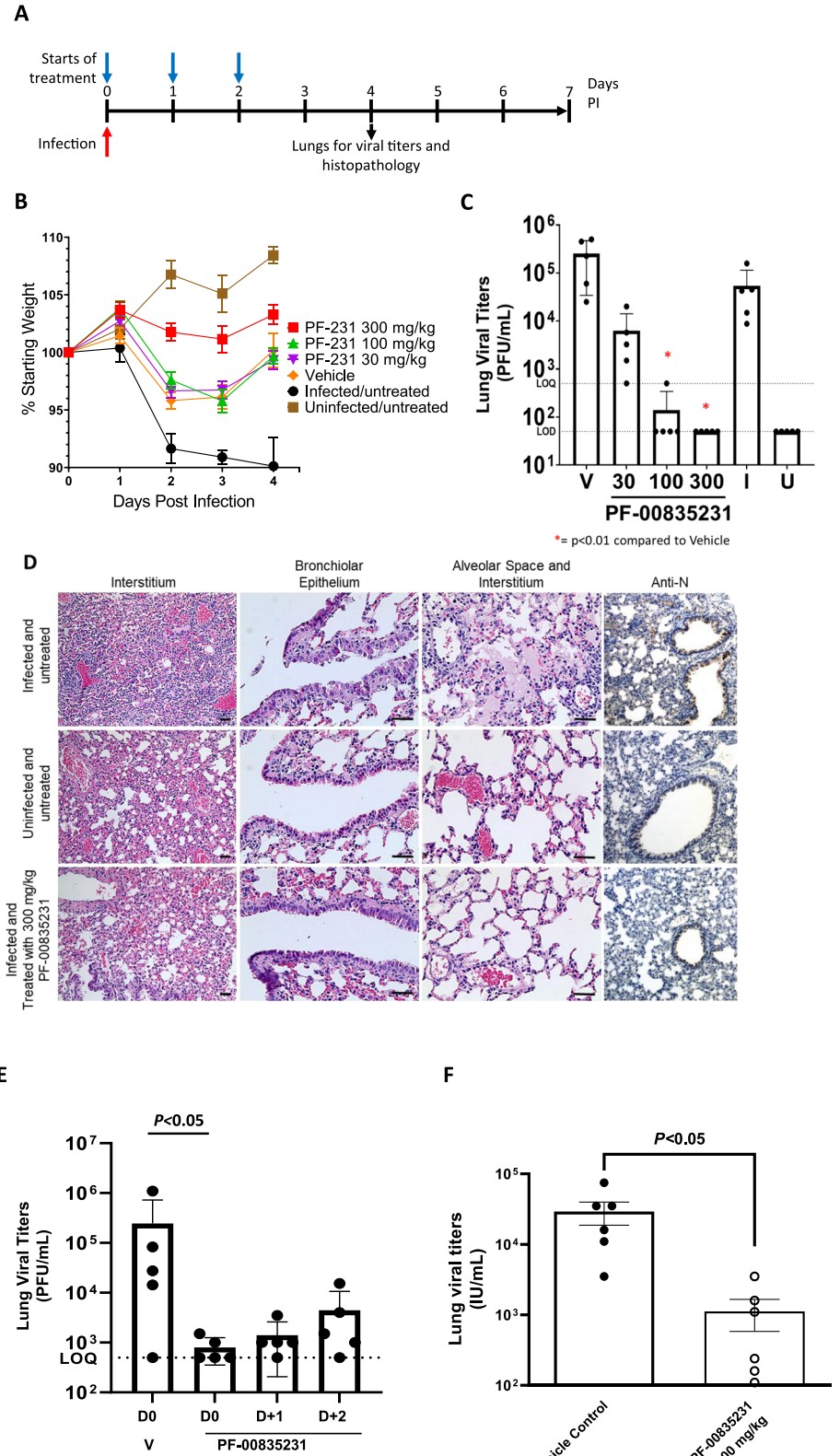

titers were determined from animals euthanized on day 4 post-infection (Fig. 4A). When the mice were treated with 30, 100, or 300 mg/kg (S.C., BID) starting on day 0, they lost less weight than infected, untreated animals (Fig. 4B). A dose-dependent reduction in lung viral titers of ≥3 $\log_{10}$ was observed (Fig. 4C). An assessment of exposure was made following 100 mg/kg in non-

infected animals and the unbound serum concentrations at $C_{min}$ were approximately 500 nM (~1x in vitro $EC_{90}$) and in the 300 mg/kg dose group the $C_{min}$ unbound exposure was 1700 nM (~3x in vitro $EC_{90}$). Microscopic evaluation of lung tissue from the animals treated in a dose–response of PF-00835231 showed that 300 mg/kg of drug prevented or decreased histopathologic

**Fig. 4 In vivo activity of PF-00835231 in mouse models of SARS-CoV infection. A** Study design for SARS-CoV-MA15 in vivo experiments. Infection with SARS-CoV-MA15 was on day 0. Treatment began on day 0 (**B–D**) or day 0, 1, or 2 post-infection (**E**). Lungs were harvested 4 days post-infection ($n = 5$ biologically independent animals per group). **B** Change in body weight of the mice starting on day 0. **C** Lung viral titers for the dose response of PF-00835231. Lung titers for treatment groups were compared to vehicle using a Kruskal–Wallis test for parts (**C**) and (**E**). Uncorrected Dunn's test was used for multiple comparisons. (p-values: 30 mg/kg= 0.15, 100 mg/kg=0.0009, 300 mg/kg=0.0003). **D** Representative photomicrographs from the dose–response experiment in (**C**) of lung sections stained with H&E (scale bar = 50 μm) or with an anti-N antibody specific for SARS-CoV (rightmost column, scale bar = 100 μm). Lungs from infected/untreated mice (top row) displayed perivascular and interstitial inflammation (top, left), degeneration and desquamation of the bronchiolar epithelium (top, middle) and proteinaceous exudate in the alveolar space with interstitial inflammatory cells (top, right), all of which were not observed in uninfected/untreated mice (middle row), or in the infected mice treated with PF-00835231 at 300 mg/kg (bottom row). IHC shows the virus within the lung tissue in the infected and untreated sample (brown stain in top right). However, treatment with PF-00835231 prevented virus from populating the lung tissue (bottom right). All images are representative of two lung sections from each of five biologically independent animals per group. **E** Lung viral titers from mice treated with PF-00835231 (100 mg/kg, BID) starting on day 0 or delayed by 1 or 2 days (p-values: D0 = 0.0151, D + 1 = 0.08, D + 2 = 0.29). **F** Mice transduced with Ad5-hACE2 were infected with SARS-CoV-2 on day 0. Treatment with PF-00835231 (100 mg/kg, BID) or vehicle started on day 0. Viral titers were determined from lungs harvested on day 3 for $n = 6$ biologically independent animals per group. Lung titers for the treatment group were compared to vehicle using a parametric, two-tailed, unpaired t-test (p-value= 0.0006). LOD, limit of detection; LOQ, limit of quantification; I, infected, untreated; U, uninfected; V, vehicle-treated; IU, international units. Viral Titers represent Mean ±SD; Weight change shown as Mean±SEM.

signs compared to vehicle-treated animals (Fig. 4D). Immunohistochemical analysis of the lung tissues clearly showed that SARS-CoV-infected cells are eliminated from the lungs of animals treated with 300 mg/kg PF-00835231 (Fig. 4D). Treatment starting at day 0 led to a significant reduction of viral titers. Importantly, delaying treatment with PF-00835231 by 1 day still resulted in a substantial reduction of viral titers (Fig. 4E).

To increase confidence in the clinical translation of PF-00835231 for COVID-19, a SARS-CoV-2 infection mouse model was also evaluated. Transduction of mice with adenovirus 5-hACE2 (Ad5-hACE2) by intranasal administration allows for infection with SARS-CoV-2[29,30]. In this model, treatment of mice with PF-00835231 at 100 mg/kg, S.C., BID, reduced lung viral titers by ≈1.5 log10 (Fig. 4F). The unbound plasma concentration at $C_{min}$ was approximately 350 nM (~0.7x in vitro $EC_{90}$), which is within a consistent 2x range to the exposure achieved with an equivalent dose in the SARS-CoV mouse model. Therefore, the viral load reduction seen in both models occurred at concentrations consistent with the in vitro antiviral activity, providing increased confidence in the predicted target exposures in the clinic.

**Favorable preclinical ADME and pharmacokinetic profile of PF-00835231.** The metabolic stability of PF-00835231 was evaluated in vitro using pooled human liver microsomes (HLM). PF-00835231 was shown to be metabolized by cytochrome P450 enzymes exhibiting an unbound $CL_{int}$ 14 μL/min/mg. Using recombinant heterologously expressed enzymes and HLM with the CYP3A selective inhibitor ketoconazole, CYP3A4 was identified as the major CYP involved in the metabolism of PF-00835231 (Table S4). It was also noted that the polymorphically expressed CYP3A5 can also metabolize PF-00835231 and that clearance may be slightly greater in CYP3A5 expressers. The potential for PF-00835231 to reversibly inhibit human cytochrome P450 enzymes (CYP1A2, 2B6, 2C8, 2C9, 2C19, 2D6, and 3A) was evaluated using probe substrates in pooled HLM and provided $IC_{50}$ values >200 μM (Table S5) and a weak signal for time-dependent inhibition (TDI) of CYP3A4/5 (Table S6). These data indicate PF-00835231 provides a low risk of causing drug-drug interactions (DDIs) on coadministration with other drugs. The potential for PF-00835231 to inhibit a range of transporters (BCRP, P-gp, OATP1B1/1B3, OCT1/2, OAT1/3 and MATE1/2 K) was evaluated using in vitro systems (Table S7). The $IC_{50}$ values were >20 μM and indicate a low risk of causing DDIs due to transporter inhibition at the projected clinical exposure. The plasma protein binding of PF-00835231 was measured across

species using equilibrium dialysis and demonstrated moderate binding to plasma proteins with plasma unbound fractions of 0.33 to 0.45 across species (Table S8).

PF-00835231 was administered IV to rats, dogs and monkeys (1 or 2 mg/kg) and exhibited moderate plasma clearances (50–80% liver blood flow), low volumes of distribution (<1.5 L/kg) and short half-lives (<2 h) across species in keeping with its lipophilic ($LogD_{7.4} = 1.7$) and neutral physiochemistry (Table S9). Following oral administration to rats (2 mg/kg) and monkeys (5 mg/kg) PF-00835231 exhibited low oral bioavailability (<2%), likely due to a combination of low absorption because of its low permeability (apparent MDCK-LE permeability of $1.3 \times 10^{-6}$ cm/s[31,32]), low solubility, potential for active efflux in the gut by P-gp and BCRP, and the potential for amide hydrolysis by digestive enzymes in the gastrointestinal tract. In rat, dog and monkey approximately 10% or less of PF-00835231 was eliminated unchanged in the urine indicating renal elimination may also play a minor role in the clearance of PF-00835231 in humans (Table S9).

**Human pharmacokinetic predictions suitable for IV administration.** Taking into account the human in vitro metabolism data and in vivo pharmacokinetic (PK) data in rats, dogs and monkeys, PF-00835231 was predicted to exhibit a plasma clearance ($CL_p$) of ~6 mL/min/kg (major CYP, minor renal pathways), steady-state volume of distribution ($Vd_{ss}$) of 1 L/kg and half-life of approximately 2 h in humans. Due to the limited oral bioavailability, short elimination half-life, and the likely need to maintain unbound systemic concentrations over time to achieve antiviral activity, a continuous IV infusion was proposed as the optimal dosing route and regimen.

**Efficacious target concentration and feasible human dose projection.** The projected minimally efficacious concentration ($C_{eff}$) was chosen to match the in vitro $EC_{90}$ (See supplement for rationale), consistent with the preclinical to clinical translation of approved protease inhibitors[33]. Since PF-00835231 was proposed to be administered by continuous infusion, the projected steady-state exposure is equal to the $C_{min}$ maintained over the dosing interval. The dose–response assay performed in the most physiologically relevant cell type, human lung carcinoma, resulted in an average $EC_{90}$ value of 0.44 μM[18]. This is consistent with additional antiviral data in Hela-ACE2 cells ($EC_{90} = 0.4$ μM) and Vero cell lines ($EC_{90} = 0.48-1.6$ μM) when a P-gp inhibitor was added to better reflect the lack of substantial P-gp transporter in

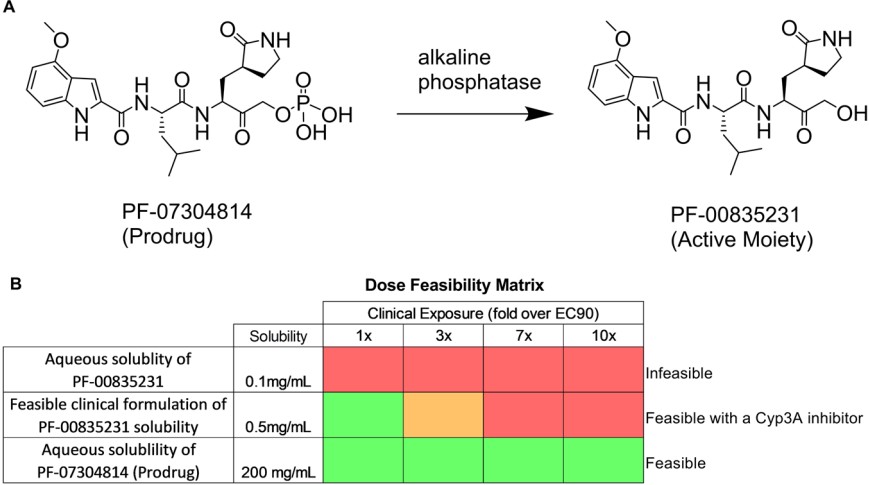

**Fig. 5 PF-07304814 prodrug and PF-00835231 structures and dose considerations. A** Chemical structure of conversion of prodrug PF-07304814 to the active moiety PF-00835231 by alkaline phosphatase. **B** Dose feasibility matrix illustrating the ability to achieve higher target exposures with increasing solubility and the limitations of dosing PF-00835231. with dosing either aqueous PF-00835231, clinically formulated PF-00835231, or aqueous PF-07304814 (prodrug). The infeasible limit (red) is assumed to be 1 L per day with a 2x potential benefit with a Cyp inhibitor (orange). Any dose under that is considered feasible (green).

the lung (Fig. 2B). Furthermore, the antiviral inhibition is supported by the antiviral time course experiment performed in a primary human airway epithelial model (preliminary data indicates an unbound $EC_{90} < 0.5\,\mu M$)[18], indicating a consistent intrinsic anti-SARS-CoV-2 activity of PF-00835231 across different cell types. Therefore, the proposed minimal $C_{eff}$ is ~0.5 $\mu M$ unbound.

Due to the rapid blood perfusion through the lungs and the continuous steady-state intravenous infusion regimen, the unbound plasma and unbound lung concentrations are assumed to be in equilibrium and, therefore, the unbound plasma concentration provides a reasonable surrogate for the concentration at the main site of action of the disease. Due to the projected short elimination half-life of PF-00835231 following an IV bolus administration, a continuous IV infusion is needed to maintain efficacious concentrations. Based on the human PK predictions, the minimally efficacious dose of PF-00835231 necessary to achieve this exposure is ~320 mg/day administered as an intravenous continuous infusion. The required duration of dosing for efficacy remains uncertain and will need to be evaluated in humans.

**Formulation and solubility profile of PF-00835231 to enable IV administration.** PF-00835231 is a moderately lipophilic ($LogD_{7.4} = 1.7$), neutral molecule with no ionizable centers throughout the physiologically relevant pH range. Consequently, PF-00835231 exhibits a pH independent solubility with an intrinsic aqueous solubility of less than 0.1 mg/mL and limited opportunities for solubility-enabling formulation approaches. Preliminary work using standard solubilizing excipients indicated that achieving a solubility >0.5 mg/mL would likely be challenging.

Based on a maximum desired intravenous infusion volume of ~1 L per day a solubility of 0.5 mg/mL would be sufficient to deliver the minimal efficacious dose estimate of ~320 mg/day to maintain a ~0.5 $\mu M$ steady-state unbound concentration (Fig. 5B). Due to the nascent understanding of the virus, the required target levels of inhibition for clinical benefit remain uncertain and the ability to evaluate exposures up to ~10x $C_{eff}$ in early clinical development is desirable. As a potential option to increase exposures, and/or decrease the required infusion volume, the use

of a strong CYP3A inhibitor (itraconazole 200 mg QD for 15 days) was considered but preliminary, physiologically based pharmacokinetic (PBPK) modeling predicted only a ~2-fold increase in PF-00835231 exposure at steady state (Table S10).

The ability to achieve higher doses could also potentially mitigate a higher than predicted clearance, or variations in patient body weight. Therefore, a medicinal chemistry strategy to significantly enhance the aqueous solubility of PF-00835231, by designing a phosphate prodrug was pursued.

**Considering an intravenous phosphate prodrug approach to improve solubility.** IV phosphate prodrugs have precedence with several commercially available drugs such as fosfluconazole and fosphenytoin which are rapidly cleaved by human alkaline phosphatase to provide high systemic exposures of their respective active moieties following IV administration (Fig. 5)[34,35]. Alkaline phosphatase is ubiquitously expressed across tissues with high levels expressed in the liver, lung, and kidney (Alkaline phosphatase tissue data from v19.proteinatlas.org[36]). High levels of conversion from prodrug to active moiety for fosfluconazole and fosphenytoin have also been observed in rats and dogs supporting cross-species translation to human for the conversion of prodrug to active moiety[35,37]. Overall, the use of a phosphate prodrug is an established approach for IV administration to provide rapid conversion to its active moiety and was considered for PF-00835231.

**Synthetic route to provide phosphate prodrug.** The synthesis of PF-00835231 has been described previously[15]. The subsequent synthesis of the phosphate prodrug of PF-00835231 was achieved via two steps (Fig. S1). Briefly, treatment of **1** (PF-00835231) with di-*tert*-butyl *N,N*-dipropan-2-ylphosphoramidite and tetrazole in tetrahydrofuran followed by oxidation with aqueous hydrogen peroxide delivered intermediate **2**. The phosphate *t*-butyl groups were subsequently hydrolyzed using trifluoroacetic acid in dichloromethane to deliver phosphate prodrug **3** (PF-07304814) as a solid.

**Enhanced formulation and solubility profile of PF-07304814 to provide clinical flexibility.** PF-07304814 rapidly undergoes in vivo conversion to the active moiety PF-00835231 (Fig. S2A).

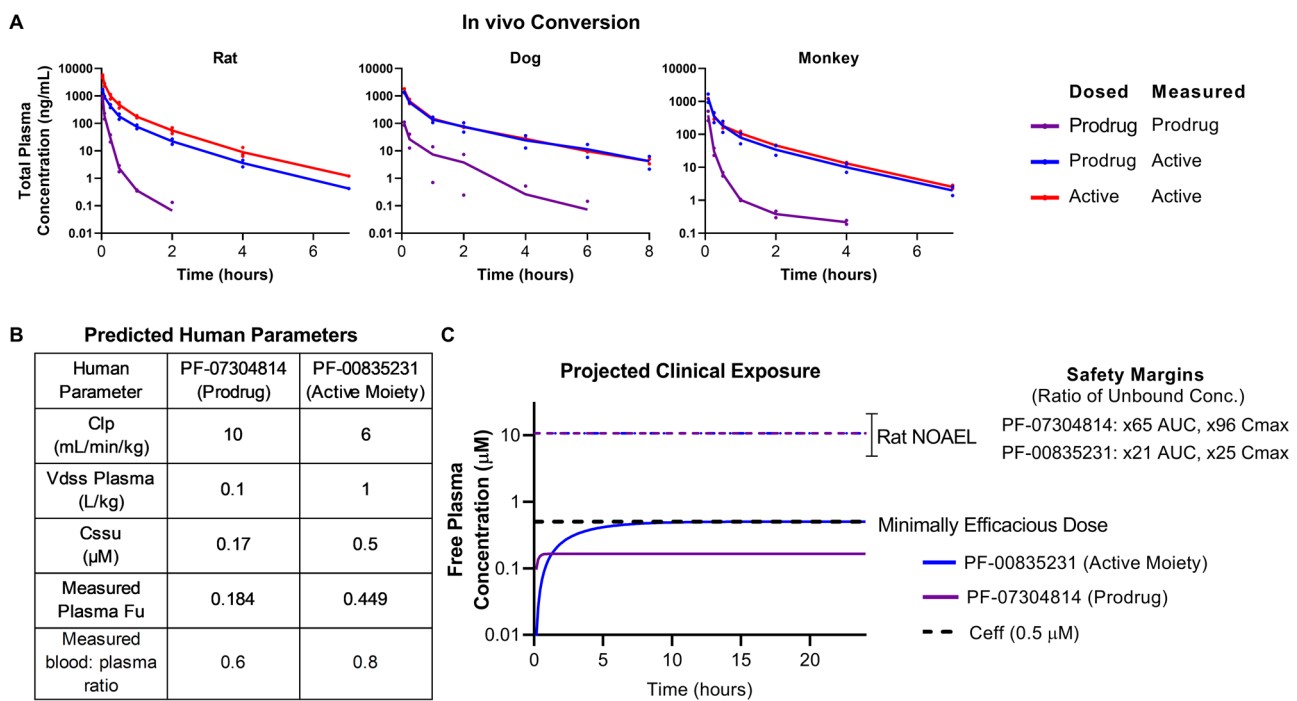

**Fig. 6 PF-07304814 (prodrug) and PF-00835231 in vivo exposure summary. A** Rat, dog and monkey PK following IV administration of PF-07304814 (1.17 mg/kg) or PF-00835231 (2 mg/kg rat, 1 mg/kg dog and monkey) demonstrating high levels of PF-00835231 formed in vivo. ($n = 2$ or 3; individual data points plotted). **B** Predicted human PK parameters and measured protein binding for PF-07304814 and PF-00835231 used for human dose prediction. **C** Projected human systemic exposure profiles at the minimally efficacious dose of 500 mg/day of PF-07304814 delivered as a continuous IV infusion. The predicted unbound steady-state concentrations for the prodrug PF-07304814 (purple) and the active moiety PF-00835231 (blue) are 0.17 μM and 0.5 μM respectively. (NOAEL = No Observed Adverse Effect Level; $C_{eff}$ = projected minimally efficacious concentration, Fu= unbound fraction; Clp= Plasma Clearance; Vdss= Volume of distribution steady state, Cssu = unbound steady-state concentration).

The phosphate prodrug is weakly acidic, with pKas of 1 and 6.4, and a predicted $LogD_{7.4}$ of −3.7. At pHs above the compound's first pKa, the phosphate functional group is de-protonated and negatively charged, which enables a significant improvement in aqueous solubility to greater than 200 mg/mL over a pH range compatible with intravenous infusion. The higher intrinsic solubility of PF-07304814 eliminates the need for solubility-enabling formulations and enables the use of standard IV compatible excipients. Furthermore, the improved solubility enables higher doses to be explored in the clinic and gives clinicians greater flexibility in terms of dose volume to account for patient-specific coadministration requirements.

**PF-07304814 (prodrug) preclinical in vitro and in vivo ADME profile.** To understand the metabolic stability and conversion of PF-07304814 to its active moiety (PF-00835231), PF-07304814 enzyme kinetics were evaluated in vitro using liver S9 fractions and was shown to exhibit rapid conversion to PF-00835231 with unbound $CL_{int}$ values of 51, 84, 168 and 428 μL/min/mg in rat, dog, monkey and human respectively. In these in vitro systems, PF-00835231 was the only metabolite formed from PF-07304814. The conversion was rapid in phosphate-free incubations but abolished in the presence of phosphate buffer supporting the role of alkaline phosphatase in this conversion (Fig. S2, Table S11). To evaluate the in vivo conversion and systemic availability of the active moiety PF-00835231, PF-07304814 was administered intravenously to rats, dogs and monkeys. PF-07304814 exhibited high conversion to PF-00835231, systemic clearance and short half-life across species forming 68, 81, 76% PF-00835231 in rats, dogs, and monkey, respectively, in comparison to the systemic exposure achieved with IV administration of PF-00835231 (Fig. 6A and Table S9).

PF-07304814 was also evaluated for the potential to cause reversible and time-dependent inhibition of human cytochrome P450 enzymes using pooled HLM and probe substrates for a range of CYP enzymes (CYP1A2, 2C8, 2C9, 2C19, 2D6, and 3A4/5) and showed low risk with $IC_{50}$ values >100 μM and no evidence of TDI (Table S5). The potential for PF-07304814 to inhibit a range of transporters (BCRP, P-gp, OATP1B1/1B3, OCT1/2, OAT1/3 and MATE1/2K) was evaluated using in vitro systems providing $IC_{50}$ values >130 μM indicating a low risk of causing DDIs due to transporter inhibition at the projected $C_{eff}$ (Table S7) The plasma protein binding of PF-07304814 was measured across species using equilibrium dialysis showing moderate binding to plasma proteins with plasma free fractions of 0.18 to 0.38 across species (Table S8).

**Encouraging human PK predictions for PF-00835231 formation.** The predicted human plasma clearance of PF-07304814 is ~10 mL/min/kg based on scaling in vitro human liver S9 $CL_{int}$ data (using Eqs. (8) and (9), see "Methods") and represents a conservative prediction of total $Cl_p$ as it only accounts for conversion of prodrug to active moiety in the liver. The human $Vd_{ss}$ for PF-07304814 is predicted to be ~0.1 L/kg based on its acidic physiochemistry and observed human $Vd_{ss}$ values of other phosphate prodrugs[34,35]. Based on the predicted $Cl_p$, $Vd_{ss}$, and a ~75% conversion to PF-00835231 based on the mean conversion in animals, PF-07304814 is anticipated to exhibit a short half-life of ~0.1 h with high conversion to the active moiety (Fig. 6B).

**PF-07304814 unlikely to contribute to antiviral activity in vivo.** In a direct comparison, using the same SARS-Cov-2 3CL^pro assay method as described in[15], the prodrug PF-07304814 binds and inhibits SARS-CoV-2 3CL^pro activity with a $K_i$ of 174 nM

providing a > 600-fold less potent $K_i$ in comparison to the active moiety PF-00835231 (0.27 nM)[15]. However, PF-07304814 shows apparently similar antiviral activity to PF-00835231 (1–12-fold, Fig. 2A) across cellular in vitro assays. This is most likely due to the partial conversion of PF-07304814 to PF-00835231 in the cellular assays by alkaline phosphatase. This was consistent with PF-00835231 concentrations measured at approximately 50% of the PF-07304814 starting concentration at the end of the 3-day incubation in the VeroE6 cell assay. It is not believed that PF-07304814 has antiviral activity without first converting to PF-00835231.

**PF-07304814 dose projection provides clinical flexibility to achieve target $C_{eff}$.** The antiviral activity and projected exposure for PF-00835231 after conversion from PF-07304814 in vivo were used to derive the minimal $C_{eff}$ and dose estimates. Based on the predicted human PK and 75% conversion of the prodrug, a free plasma concentration of the active moiety PF-00835231 of 0.5 µM ($C_{eff}$) can be achieved with a 500 mg continuous IV infusion of the prodrug over 24 h (Fig. 6C). The estimated time to achieve 90% steady-state exposure of PF-00835231 is approximately 6 h. Due to the improved solubility (>200 mg/mL), the dose of PF-07304814 can be delivered in a volume of less than 0.25 L. In addition, the dose can be increased if the observed human plasma Cl exceeds 6 mL/min/kg, if the percent converted from prodrug to active is less than predicted, or if exposures in excess of the minimal $C_{eff}$ (0.5 µM free) are required to maximize clinical activity (Fig. 5B). Overall, the improved solubility of PF-07304814 would theoretically enable >100-fold the proposed minimal $C_{eff}$ dose in a 0.25 L dose volume.

**Preclinical safety profile supports progression to clinical evaluation.** A toxicology assessment consisting of an in vitro battery of genetic toxicity, secondary and safety pharmacology studies, in conjunction with a single species (rat) in vivo Good Laboratory Practice (GLP) study has been completed.

The safety profiles of PF-07304814 and PF-00835231 were assessed individually in a range of in vitro and in vivo safety studies in rats. In the in vitro studies, PF-07304814 and PF-00835231 were negative in the bacterial reverse mutation assay and did not induce micronuclei formation. Both compounds had minimal potential for secondary (off-target) pharmacology at clinically relevant exposures. Neither PF-07304814 nor PF-00835231 inhibited hERG current amplitude at up to 300 µM (1,770- and 600-fold, respectively, in reference to the projected unbound human $C_{max}$ of 0.17 and 0.50 µM, respectively, at the projected human efficacious dose), indicating a favorable cardiovascular safety profile. In human blood hemocompatibility assays, both compounds had no effect on hemolysis or flocculation/turbidity parameters, indicating compatibility with human blood and supporting intravenous administration.

PF-07304814 was administered to rats via continuous IV infusion for 24 h in a GLP study. There was no test article related findings and no target organ toxicity was identified. PF-07304814 had no effects on neurological safety pharmacology parameters as assessed by functional observation battery in the 24-h continuous IV infusion rat study. The no observed adverse effect level (NOAEL) was 1000 mg/kg. PF-00835231 was also administered to male rats via continuous IV infusion for 4 days in a non-GLP exploratory toxicity study and was tolerated at 246 mg/kg/day, the highest feasible dose tested. PF-00835231-related findings in this study were limited to minimal, non-adverse effects on clinical chemistry parameters including higher mean triglycerides (1.9-3.6x vs controls), cholesterol (1.3x), and phosphorus (1.1x) without any microscopic correlates or associated functional

changes. No test article related adverse effects were seen in any study.

At the NOAEL from the 24 h, GLP continuous IV infusion study with PF-07304814 in rats, the anticipated exposure margins for unbound $C_{max}$ and $AUC_{24}$ are 97 and 65-fold for PF-07304814 and 25 and 21-fold for PF-00835231, at the projected minimum human efficacious dose of 500 mg/day. This indicates the potential to safely evaluate multiples over $EC_{90}$ in humans during clinical testing to understand the exposure response relationship and to achieve high levels of inhibition, if required. Furthermore, no overlapping or additive toxicity with medications currently being used in standard of care COVID-19 treatment is expected with administration of PF-07304814 in humans, making PF-07304814 an attractive combination partner. Based on results from the set of safety studies conducted, PF-07304814 exhibited an encouraging nonclinical safety profile and supported progression into Phase 1 clinical studies.

## Discussion

PF-07304814 is a phosphate prodrug that is rapidly converted in vivo to the active moiety, PF-00835231, which exhibits high selectivity over human proteases, acts as a broad-spectrum coronavirus 3CL protease inhibitor, and demonstrates potent antiviral activity in vivo. Robust antiviral activity was demonstrated in a range of cellular in vitro assays in keeping with SARS-COV-2 human airway epithelial data[18] suggesting a $C_{eff}$ value of ~0.5 µM unbound. This $C_{eff}$ exposure was consistent with the free plasma concentration associated with the maximal viral load decrease in the in vivo mouse SARS-CoV model and a significant reduction in viral load in a SARS-CoV-2 model as well. To date resistance studies exploring how SARS-CoV-2 may attempt to circumvent a targeted 3CL^pro therapy have not been performed but would be informative. The predicted human pharmacokinetics of PF-07304814 provides the ability to achieve systemic unbound concentrations of 0.5 µM ($EC_{90}$) of PF-00835231 by delivering 500 mg as a continuous infusion over 24 h with infusion volumes of less than 0.25 L. In addition, higher doses (up to and beyond 10x $C_{eff}$) also remain feasible due to the high solubility of PF-07304814.

Overall, PF-07304814 exhibits an encouraging preclinical profile that has the ADME, safety, and once converted to PF-00835231, SARS-CoV-2 antiviral activity to support progression to the clinic as a COVID-19 single-agent antiviral treatment, with potential for further additional benefit in combination with antivirals that target other critical stages of the coronavirus life cycle. The favorable profile of PF-07304814 supported the progression to clinical trials in healthy volunteers and COVID-19 patients (NCT04627532 and NCT04535167).

## Methods

**Thermal stability assay.** 3CL^pro protein (3 µM) was incubated with DMSO or 40 µM PF-00835231 in reaction buffer (20 mM HEPES, pH 6.5, 120 mM NaCl, 0.4 mM EDTA, 4 mM DTT and 20% glycerol) at 30 °C for 30 min. 1X SYPRO orange dye was added and fluorescence of the well was monitored under a temperature gradient range from 30 °C to 90 °C with 0.05 °C/s incremental step. The melting temperature ($T_m$) was calculated as the mid-log of the transition phase from the native to the denatured protein using a Boltzmann model in Protein Thermal Shift Software v1.3.

**Coronavirus protease panel.** Expression and purification of Alpha-, Beta-, and Gamma-coronavirus 3 CL^pro enzymes: In general, the coding regions for all of the 3C-like proteases described in this study were codon-optimized and synthesized by a commercial source (BioBasic or GeneScript). The coding regions were inserted into a derivative of the pET 11 or pET15 expression vectors that place a hexa-histidine tag at the N-terminus that is followed by a Tobacco-Etch Virus (TEV) protease cleavage site that resides between the coding region of 3CL^pro and the hexa-histidine affinity tag. The cleavage site for TEV is also recognized by 3CL^pro, and therefore it is

autocatalytically cleaved during expression in *Escherichia coli*, releasing authentic 3CL$^{pro}$ for purification. Each of the 14 coronaviruses were expressed in either *E. coli* BL21(DE3) or *E. coli* BL21(DE3) Gold cells and then purified after lysis using a combination of ammonium-sulfate fractionation, hydrophobic-interaction, anion-exchange, cation-exchange and size-exclusion chromatography.. Additional details for purification of

SARS-CoV-2 3CL$^{pro}$[38] SARS-CoV[39,40], MERS[41], HKU4[42], HKU5[43], FIPV[44], PEDV[45], MHV[46] and the other 3CL pro enzymes[47] based on the construct in Fig. S3 have been published. Expression and purification protocols for 3CL$^{pro}$ from NL63, OC43, HKU9 and IBV coronaviruses were performed as described[47] with the following modifications. NL63 3CL$^{pro}$ was expressed and purified using a modified protocol based on the purification of murine hepatitis virus (MHV) 3CL$^{pro}$ using a DEAE-Cellulose column. OC43 and HKU9 3CL$^{pro}$ were expressed using the autoinduction protocols developed for MERS 3CL$^{pro}$ followed by the same purification method as MHV 3CL$^{pro}$. Avian coronavirus (IBV) 3CL$^{pro}$ was expressed and purified using a modified protocol based on MERS 3CL$^{pro}$.

Expression and purification of the human h229E coronavirus 3CL$^{pro}$ were performed using the following method. The sequence for human h229E was obtained from UniProt (accession AGW80931.1) and was designed to include an N-terminal hexa-histidine tag followed by polyprotein residues 2960 - 3267. This gene sequence was codon-optimized for expression in *E. coli* and subcloned into a pET11a vector using the synthetic DNA services of GenScript Biotech (Piscataway, NJ). The pET11a plasmid containing the 229E 3CL$^{pro}$ gene was electroporated into *E. coli* BL21-GOLD (DE3) cells. A single colony of the transformed cells was used to inoculate 100 mL of 2xYT media (10 g yeast extract, 16 g tryptone, 5 g NaCl per 1 L of water, 100 µg/mL carbenicillin, pH adjusted to 7.50 using 10 M NaOH). The preculture was incubated at 37 °C until the culture reached an OD$_{600}$ of 0.6. 15 mL of the preculture was added to 1 L of 2xYT expression media and incubated at 37 °C until an OD$_{600}$ of 0.6 was reached. The cultures were then placed on ice for 15 min before the addition of isopropyl B-D-1-thiogalactopyranoside (IPTG) to a final concentration of 0.6 mM and further incubated at 25 °C. After 16 h, the cells were harvested by centrifugation at $10,940 \times g$ for 10 min to yield an 8.3 g cell pellet per L. The cell pellet was resuspended in 5 mL Lysis Buffer (25 mM HEPES, 0.05 mM EDTA, 5 mM β-mercaptoethanol (β-ME), 1 mg/mL lysozyme) per 1 g of pelleted cells using a manual homogenizer. The homogenized cell suspension was sonicated for a total of 12 min at an amplitude of 60% for periods of 10 s with 20 s delays using a Branson digital sonifier. Solid ammonium sulfate was slowly added to the resulting lysate to a final concentration of 1 M and stirred at 25 °C for 10 min. The lysate was then clarified via centrifugation at $20,442 \times g$ for 45 min at 4 °C.

The resulting supernatant was checked for activity prior to being applied to a 75 mL Phenyl Sepharose (GE Healthcare) resin equilibrated in Buffer B (1 M ammonium sulfate, 50 mM HEPES, pH 7.50, 0.05 mM EDTA, 5 mM βME). The protein was eluted off the column using a linear gradient to 100% Buffer A (25 mM HEPES, 0.05 mM EDTA, 5 mM β-ME) over 5 column volumes, followed by a water wash. 2 mL fractions that eluted in water containing 229E 3CL$^{pro}$ were mixed with 8 mL of 5X Buffer A. Fractions were assessed for purity via SDS-PAGE and enzymatic activity based on specific activity prior to pooling. The resulting protein pool was filtered using a 0.45 µm surfactant-free cellulose acetate membrane prior to injection onto an 8 mL MonoQ column equilibrated in Buffer A. The protein was eluted using a linear gradient to 100% Buffer C (1 M NaCl, 50 mM HEPES, pH 7.50, 0.05 mM EDTA, 5 mM βME) over 15 column volumes. Fractions were pooled based on purity and specific activity and concentrated to 2.5 mL using a 10,000 molecular weight cutoff (MWCO) spin concentrator (MilliporeSigma). The sample was then injected onto an SRT SEC-300 column (Sepax Technologies) that had been equilibrated in Buffer D (50 mM HEPES, pH 7.50, 10% glycerol, 2.5 mM dithiothreitol (DTT)). The resulting protein was pooled based on the criteria above, aliquoted and flash-frozen in liquid nitrogen, and stored at −80 °C.

**Inhibitor characterization of 3CL$^{pro}$ enzymes**. Inhibition of all 3CL$^{pro}$ enzymes was measured using a continuous, FRET assay. The increase in fluorescence due to the cleavage of a custom-synthesized substrate, UIVT3 (HiLyte Fluor$_{488}$$^{TM}$-ESATLQSGLRKAK-QXL$_{520}$$^{TM}$-NH$_2$) (Anaspec) by the enzymes was monitored over time. All $K_i$ determinations were performed in Costar 3694 EIA/RIA 96-well, half-area, flat bottom, black polystyrene plates (Corning) using an assay buffer that contained 50 mM HEPES (pH 7.5), 0.1 mg/mL bovine serum albumin (BSA),

0.01% Triton X-100 and 2 mM DTT. The final assay volume was 100 µL and all assays were performed in triplicate. 1 µL of 100X inhibitor stocks, prepared in 100% DMSO, were added to assay buffer and incubated for 5 min at 25 °C. Then, 3CL$^{pro}$ enzyme was added to the mixture and incubated for 10 min at 25 °C. The final concentrations of each coronaviral 3CL$^{pro}$ enzyme are listed below and they are different depending on the coronavirus. All reactions were initiated with the addition of UIVT3 substrate to a final concentration of 2 µM.

The fluorescence of the substrate (excitation 485 nm/emission 528 nm) was measured using either a CLARIOstar Plate Reader (BMG Labtech) or Synergy H1 hybrid multi-mode plate reader (Biotek). The initial rates at each inhibitor concentration were obtained by dividing the Relative Fluorescence Units (RFU) produced during the initial rate period of the enzyme by time in minutes, yielding RFU min$^{-1}$. Initial velocities were calculated by dividing the observed velocity in the absence of inhibitor (V$_o$) by the initial velocity at different inhibitor concentrations (V$_i$). $K_i$ values, with the exception of MERS 3CL$^{pro}$, were obtained by fitting the data to the Morrison equation (Eq. (1)) for competitive inhibition[48].

$$\frac{V_i}{V_o} = \frac{[E]_o + [I]_o - K_i(1 + \frac{[S]}{K_m})}{2[E]_o} + \frac{\sqrt{\left([I]_o + K_i\left(1 + \frac{[S]}{K_m}\right) - [E]_o\right)^2 + 4[E]_o K_i\left(1 + \frac{[S]}{K_m}\right)}}{2[E]_o} \quad (1)$$

For these fits, the $V_{max}$ was initially set to a value less than 1.2, the UIVT3 substrate concentration was fixed at 2 µM, and the $K_m$ value was fixed at a value of 250 µM. The $K_m$ value for the UIVT3 substrate is estimated to be higher as even after correcting for inner-filter effects via dilution experiments, the 3CL$^{pro}$ enzymes were not saturated with substrates as the response of the enzyme to increasing UIVT3 substrate concentrations was linear over the range of 1–250 µM.

The total enzyme concentrations were experimentally determined by measuring their absorbance at 280 nm and using their respective molar extinction coefficients ε$_{280}$ calculated based on their primary amino acid sequences. The values for enzyme concentration $[E]_o$ in Eq. (1) were not fixed during the curve-fitting process. The resulting enzyme concentrations from the Morrison equation fit are represented in the parentheses following the experimentally determined total enzyme concentrations as follows: 200 nM SARS-2 (188 ± 10 nM), 200 nM SARS (153 ± 2 nM), 1000 nM MERS, 250 nM NL63 (170 ± 4 nM), 125 nM HKU1 (59 ± 2 nM), 100 nM HKU4 (60 ± 1 nM), 125 nM HKU5 (74 ± 1 nM), 400 nM HKU9 (265 ± 5 nM), 200 nM 229E (116 ± 3 nM), 100 nM OC43 (51 ± 1 nM), 100 nM PEDV (40 ± 1 nM), 100 nM MHV (75 ± 4 nM), 25 nM FIPV (36 ± 1 nM), 50 nM IBV (31 ± 1 nM). MERS 3CL$^{pro}$ data were calculated using Eq. (2) and fit to the substrate inhibition equation. %$V_{max}$ is the maximum percent activity observed. All fits were performed using GraphPad Prism 8.3.

$$\%\text{Rate} = \frac{\%V_{max} * [\text{Inhibitor}]}{K_{a,app} + [\text{Inhibitor}] * (1 + \frac{[\text{Inhibitor}]}{K_{i\,app}})} \quad (2)$$

**Mammalian protease panel**. The respective protease in assay buffer (50 mM Tris with 100 mM sodium chloride and Brij 35 at pH = 8 except for cathepsin D pH = 3.5 and HIV pH = 5.5) was added to assay ready compound plates. The enzymatic reaction was initiated with the addition of indicated substrate in assay buffer. Final concentrations of respective protease and substrate are shown in Table 2. Initial rates were measured by following the fluorescence of the cleaved substrate (Ex/Em 355/460 nm) using a Spectramax (Molecular Devices) fluorescence plate reader in the kinetic format.

Percent inhibition values were calculated based on control wells containing no compound (0% inhibition) and wells containing a control compound (100% inhibition). IC$_{50}$ values were generated based on a four-parameter logistic fit model using ActivityBase software (IDBS). Percent activity values were calculated based on control wells containing no compound (100% activity) and wells containing a control compound (0% activity).

**Antiviral activity**. The ability of compounds to inhibit viral-induced cytopathic effect (CPE) against human coronaviruses (SARS-CoV, SARS-CoV-2, hCoV-229E) was assessed by monitoring cell viability using two different assay endpoints in VeroE6 or MRC-5 cells. VeroE6 cells that are enriched for hACE2 expression were batched inoculated with SARS-CoV-2 (USA_WA1/2020) at a multiplicity of infection (MOI) of 0.002 in a BSL-3 lab (Southern Research Institute). Virus

**Table 2 Human and HIV protease and substrate concentrations.**

| Protease/class | Enzyme (nM) | Substrate | Substrate (µM) |
|---|---|---|---|
| Caspase 2/ cysteine | 10 | Ac-LEHD-AMC | 5 |
| Cathepsin B/ aspartyl | 1.2 | CBZ-Arg-Arg-AMC | 15 |
| Cathepsin D/ aspartyl | 1.0 | MCA-PLGL-Dap(Dnp)-AR-NH2 | 2 |
| Chymotrypsin/ serine | 0.5 | Suc-AAPF-AMC | 10 |
| Elastase/ serine | 0.6 | MeOSuc-AAPV-AMC | 10 |
| HIV-1/ aspartyl | 20 | Anaspec SensoLyte | 10 |
| Thrombin a/ serine | 0.01 | H-D-CHA-Ala-Arg-AMC.2AcOH | 10 |

inoculated cells are then added to assay ready compound plates at a density of 4000 cells/well in DMEM containing 2% heat inactivated eFBS. Following a 3-day incubation at 37 °C with 5% $CO_2$, a time at which virus-induced cytopathic effect is 95% in the untreated, infected control conditions. Cell viability was evaluated using Cell Titer-Glo (Promega), according to the manufacturer's protocol, which quantitates ATP levels. Cytotoxicity of the compounds was assessed in parallel in assay ready compound plates with non-infected cells.

VeroE6-EGFP cells were seeded at a density of 2000 cells/well in DMEM containing 2% FCS and 0.08% sodium bicarbonate were seeded into 384-well cell culture microplates containing the serially diluted test compounds. Immediately following cell seeding, virus (SARS-CoV-2 (Belgium strain) or SARS-CoV (CDC 200300592)) was added at a MOI of 0.015 equaling ~30 PFU in a Caps-It isolator system (Rega Institute). The plates were then automatically transferred to an integrated incubator and incubated for 5 days at 37 °C with 5% $CO_2$, a time at which virus-induced cytopathic effect is 100% in the untreated, infected control conditions (virus controls). Uninfected assay ready compound plates were treated the same way and were used to measure compound cytotoxicity. To monitor the EGFP fluorescence, the plates were transferred to a high-content imager for fluorescence microscopy. A ×5 objective was used to capture one field per well (approximately 70% of the well). The images were captured with one channel on auto-focus, a fixed exposure time of 0.023 s and with a fixed objective offset (z offset). For the image analysis, the SpotDetector bio-application from the Cellomics software was used, which requires two (2) channels, one (1) for object identification (first channel) and one (1) for signal/spots identification (second channel). The Valid Object Count reported value was the total count of the number of nuclei.

MRC-5 cells, seeded at a density of 20,000 cells/well were incubated overnight in MEM containing 5% FBS at 37 °C and 5% $CO_2$. (Wuxi AppTech). The following day test compounds, HCoV-229E virus (ATCC VR-740) (200 TCID50) was added at concentrations which correspond to a multiplicity of infection (MOI) of 0.007, were added to the MRC-5 cells. Cells were incubated for 3 days at 35 °C with 5% $CO_2$ and cell viability was evaluated using the CellTiter Glo (Promega) post the 3-day incubation, according to the manufacturer's protocol. Cytotoxicity of test compounds was assessed in parallel with cells plated 1 day prior to addition of compound, but incubated for 5 days with compound only, and then treated with CellTiter Glo (Promega).

Test compound(s) were tested either alone or in the presence of the P glycoprotein (P-gp) inhibitor, CP-100356 at indicated concentrations of either 0.5 or 2 µM. The inclusion of CP 100356 was to assess if the test compound(s) were being effluxed out of cells due to the expression of P-glycoprotein. The activity of CP-100356 by itself was assessed with an $EC_{50}$ of 23.4 µM and $CC_{50}$ of 29.6 µM. These values are an order of magnitude higher than those tested in combination with PF-00835231. The $EC_{50}$ values for PF-00835231, in combination with ascending concentrations of CP-100346: 0, 0.5, 1.0, and 2.0 µM were 35.9 µM, 2.36 µM, 0.95 µM and 0.46 µM, respectively, with all resulting in CC50 values of >50 µM. The TI for PF-00835231 in combination with 0, 0.5, 1.0, and 2.0 µM of the P-gp inhibitor, in the VeroE6-SARS-CoV2 CPE assay, were 1.39, 21.2, 52.6, and 109, respectively. These data demonstrate negligible antiviral activity of our P-gp inhibitor, CP-100356 with an $EC_{50}$ of 23.4 µM. Cytotoxicity was demonstrated with an $CC_{50}$ of 29.6 uM. In the presence of 2 µM inhibitor, the potency of PF-00835231 increased 78-fold without cytotoxicity observed up to 50 µM. In summary, at the doses tested, CP-100356 is not contributing to either antiviral or cytotoxicity activity of PF-00835231.

The percent effect at each concentration of test compound was calculated based on the values for the no virus control wells and virus-containing control wells on each assay plate. The concentration required for a 50% response ($EC_{50}$) value was determined from these data using a 4-parameter logistic model. $EC_{50}$ curves were fit to a Hill slope of 3 when >3 and the top dose achieved ≥ 50% effect. If cytotoxicity was detected at greater than 30% effect, the corresponding concentration data was eliminated from the $EC_{50}$ determination. For cytotoxicity plates, a percent effect at each concentration of test compound was calculated based on the values for the cell only control wells and hyamine or no cell containing control wells on each assay plate. The $CC_{50}$ value was calculated using a 4-parameter logistic model. A therapeutic index (TI) was then calculated by dividing the $CC_{50}$ value by the $EC_{50}$ value.

Drug combination studies were performed using HeLa-ACE2 cells in a high-content imaging assay. HeLa-ACE2 cells were seeded at a density of $1.0 \times 10^3$ cells per well in DMEM containing 2% FBS into the 384-well µclear-bottom assay ready compound plates. Plated cells were transferred to the BSL-3 facility where 13 µL of SARS-CoV-2 (strain USA-WA1/2020, propagated in Vero E6 cells) diluted in DMEM with 2% FBS was added per well at a concentration to achieve ~30–50% infected cells. Plates were incubated for 24 h at 34 °C with 5% $CO_2$, and then fixed with 25 µL of 8% paraformaldehyde for 1 h at 34 °C with 5% $CO_2$. Plated cells were transferred to the BSL-3 facility where SARS-CoV-2 (strain USA-WA1/2020) was added per well at a concentration to achieve ~30–50% infected cells. Plates were incubated for 24 h at 34 °C with 5% $CO_2$, and then fixed with 25 µL of 8% paraformaldehyde for 1 h at 34 °C with 5% $CO_2$. Plates were washed with 1X phosphate buffered saline (PBS) containing 0.05% Tween 20 after fixation, primary and secondary antibody staining. Human polyclonal sera, from two different recovered patients, was diluted 1:500 in Perm/Wash buffer, added to the plate and incubated at room temperature for 2 h for primary staining. De-identified sera was kindly provided through the "Collection of Biospecimens from Persons Under

Investigation for 2019-Novel Coronavirus Infection to Understand Viral Shedding and Immune Response Study" UCSD IRB# 200236. Protocol was approved by the UCSD Human Research Protection Program. Informed consent was obtained. The two patient sera were chosen based on the brightness and specificity of the staining of infected cells, and therefore for their utility as detection reagents. Two different sera were used to confirm that we did not observe serum-specific detection artefacts in our assay. 6 µg/mL of goat anti-human H+L conjugated Alexa 488 together with 8 µM of antifade-4,6-diamidino-2-phenylindole dihydrochloride (DAPI) in SuperBlock T20 (PBS) blocking buffer was added and incubated at room temperature for 1 h in the dark for secondary staining. Plates were imaged using the ImageXpress Micro Confocal High-Content Imaging System (Molecular Devices) with a ×10 objective, with four fields imaged per well. Images were analyzed using the Multi-Wavelength Cell Scoring Application Module (MetaXpress). The total number of cells were determined with DAPI staining to identify the host-cell nuclei and the number of SARS-CoV-2 infected cells were determined by the SARS-CoV-2 immunofluorescence.

Data were analyzed using Genedata Screener, Version 17.0.1-Standard. Primary in vitro screen and the host-cell toxicity screen data [% Positive W2 (MultiWaveScoring)] were uploaded to Genedata Screener. For $EC_{50}$ determinations, data were normalized to negative inhibition controls (DMSO) minus positive viral inhibitor controls (2.5 µM remdesivir) and for host-cell cytotoxicity in the viral infected cells (concentration required for 50% cytotoxicity; $CC_{50}$), data were normalized to negative cytotoxic controls (DMSO) minus positive cytotoxic controls (10 µM puromycin). The "% Positive W2 (MultiWaveScoring)" and "Total Cells (MultiWaveScoring)" values were used for analysis of antiviral effect and host-cell toxicity, respectively. Compounds were tested in technical triplicates on different assay plates and dose curves were fitted with the four-parameter Hill Equation. Replicate data were analyzed using median condensing. GeneData Screener was used to assess drug combination effects, at 90% effect, using the Loewe, Bliss, and HSA models or SynergyFinder using the ZIP model[49].

In general, a synergy score of >1 and a combination index of <1 indicate that the combination treatment has a synergistic effect[23]. To assess whether synergy could be achieved at high inhibition levels, the isobologram level was set at 0.9 to capture meaningful synergy with a 90% viral reduction (equivalent to a $1\log_{10}$ reduction).

**In vivo efficacy studies**. Animal treatment and subsequent observations of animal weight, and virus titration were conducted in the BSL-3 laboratory at University of Maryland College Park under IACUC approved protocols. BALB/c mice housed in the BSL-3 were anesthetized with ketamine/xylazine mixture and intranasally inoculated with $1 \times 10^5$ pfu of MA15-SARS-CoV in 50 µL of PBS total volume. In all experiments the vehicle for PF-00835231 was 0.05% (v/v) polysorbate-80, 0.1% (w/v) sodium carboxymethylcellulose, dosed at 10 mL/kg S.C. BID. While on the study, mice were weighed daily and observed for clinical signs of disease. At day 4 post-infection, 5 mice per group were euthanized by isoflurane inhalation and lung lobes were either fixed in 4% paraformaldehyde (PFA) at 4 °C or placed in 1 mL PBS with sterile glass beads at −80 °C.

For histopathology, lungs in 4% PFA were removed from the BSL3 at least 24 h after they were added to PFA and embedded in paraffin for sectioning and H&E staining at the University of Maryland at Baltimore Histology Core. Slides were read by Norimitsu Shirai in the Pfizer Groton, DSRD Global Pathology group.

MA15-SARS-CoV lung titers were quantified by homogenizing mouse lungs in 1 ml phosphate-buffered saline (PBS) using 1.0 mm glass beads and a Beadruptor. VeroE6 cells are plated in 6 well plates with $1 \times 10^5$ cells per well. MA15-SARS-CoV virus titer in plaque-forming units was determined by plaque assay: 25 µL of the lung homogenate was added to 225 µL of PBS and diluted 10-fold across a 6-point dilution curve with 200 µL of diluent added to each well. After 1 h, a 3 mL agar overlay containing MEM was added to each well. Plates were incubated for 3 days at 37 °C (5% $CO_2$) before plaques were counted. Lung titers for the treatment groups were compared to vehicle and statistically analyzed using a non-parametric one-way ANOVA with Kruskal-Wallis test using GraphPad Prism v. 9.0.0. Uncorrected Dunn's test was used for multiple comparisons of treatment groups to vehicle group.

For immunohistochemistry, 5 µm sections were cut from paraffin-embedded blocks and placed onto positive charged slides. The sections were immunostained using the Dako EnVision FLEX + detection system (DAKO, Carpinteria, CA). The Dako PT link was used for deparaffinization and heat-induced epitope retrieval using Dako Target Retrieval Solution, High pH, for 20 min. Endogenous peroxidase activity was blocked with DAKO Peroxidase-Blocking Reagent for 5 min before incubation with rabbit polyclonal SARS Nucleocapsid Protein Antibody [NB100-56576] (Novus Biologicals, Centennial, CO), 1:400 for 20 min at room temperature followed by DAKO Anti-rabbit HRP Detection Reagent for 20 min. Finally, the sections were incubated 10 min with DAKO diaminobenzene (DAB) before they were counterstained with Dako FLEX hematoxylin, rinsed and mounted in Cytoseal XYL (Thermo Scientific, Waltham, MA).

Animal studies using the Ad5-hACE2 mouse model were performed in animal biosafety level 3 (BSL3) facility at the Icahn school of Medicine in Mount Sinai Hospital, New York City. All work was conducted under protocols approved by the Institutional Animal Care and Use Committee (IACUC). We utilized female 8-week-old specific pathogen–free BALB/c mice (the Jackson laboratory strain

000651). Five days prior to infection with SARS-CoV-2, BALB/c mice were infected intranasally with $2.5 \times 10^8$ PFU of an adenovirus carrying the gene for hACE2. Viral seed stocks for non-replicating E1/E3 deleted viral vectors based on human adenovirus type-5 expressing the human angiotensin-converting enzyme 2 (Ad-ACE2) receptor under the control of a CMV promoter, were obtained from the Iowa Viral Vector Core Facility. Viral stocks were amplified to high titers following infection of T-RexTM-293 cells and purification using two sequential rounds of cesium chloride (CsCl) ultracentrifugation, as described previously[50,51]. The infectious titer was determined using a tissue culture infectious dose-50 (TCID$_{50}$) end-point dilution assay, and physical particle titer quantified by micro-bicinchoninic acid (microBCA) protein assay. Both are described in more detail previously[50].

Mice were anesthetized with a mixture of ketamine/xylazine before each intranasal infection with SARS-CoV-2 USA-WA1/2020. 3 days post-infection (dpi) animals were humanely euthanized. Whole lungs were harvested and homogenized in PBS with silica glass beads then frozen at $-80\,^{\circ}$C for viral titration via TCID$_{50}$. Briefly, infectious supernatants were collected at 48 h post-infection and frozen at $-80\,^{\circ}$C until later use. Infectious titers were quantified by limiting dilution titration using Vero E6 cells. Briefly, Vero E6 cells were seeded in 96-well plates at 20,000 cells/well. The next day, SARS-CoV-2-containing supernatant was applied at serial 10-fold dilutions ranging from $10^{-1}$ to $10^{-6}$ and, after 5 days, viral cytopathic effect (CPE) was detected by staining cell monolayers with crystal violet. TCID$_{50}$/ml were calculated using the method of Reed and Muench. The Prism software (GraphPad v 9.0.0) was used to determine differences in lung titers using a parametric, two-tailed, unpaired t-test.

To evaluate exposure in the in vivo efficacy studies, blood samples (approximately 35ul) were sampled from the mouse tail vein and blood, plasma or serum was prepared and stored frozen at $-20\,^{\circ}$C or lower. For analysis samples were protein precipitated with acetonitrile:methanol containing an internal standard propranolol (50 ng/ml) and concentrations were quantified via HPLC-MS against a calibration curve. For further details see the preclinical pharmacokinetic studies and bioanalytical LC-MS analysis section.

**LogD, solubility and pKa measurements**. The LogD of PF-00835231 was measured at pH 7.4 using a shake-flask method dissolving PF-00835231 in 1-octanol, mixing with 0.1 M sodium phosphate buffer to determine the partition coefficient between the aqueous and organic phase[52]. 2 μL of a 10 mM DMSO stock of PF-00835231 was added to a 96 well in duplicate containing Buffer-saturated 1-octanol (149 μL/well) and 1-octanol saturated phosphate buffer (149 μL/well). Plates were sealed with silicone well cap-mats and vigorously mixed on their sides for 1 h at a speed of 10 at room temperature on a plate shaker then subjected to centrifugation at 2500 rpm ($1006 \times g$) for 15 min. After the cap-mats were removed from the plates, aliquots (2 μL/well) from the 1-octanol phase were transferred into new 1 mL capacity plates containing 1-octanol IS solution (398 μL/well). Aliquots (10 μL/well) from the buffer phase were transferred into other 1 mL capacity plates containing buffer IS solution (190 μL/well). 20 μl samples were analysed by high-performance liquid chromatography-tandem mass spectrometry (HPLC-MS). Peak areas were corrected by dilution factors and incorporating and the ratio of the peak areas were used to calculate the results.

The LogD of PF-07304814 was predicted using ACDlabs (v2019.1.1). Solubility was measured at room temperature by slurrying the solid drug substance in deionized water and pH buffers for 48 h, passing the slurry through a 0.2 μm filter, and analyzing the filtrate using LC-UV spectrometry. Solution state NMR was used to measure the chemical shift of proton and phosphorous signals as a function of pH, and the pKa was then calculated based upon a two-site binding model[53]. Both $^1$H and $^{31}$P NMR data were collected on 600 MHz spectrometers at 298 K, using a Bruker-Biospin 5 mm TCI cryoprobe or a 5 mm BBFO cryoprobe respectively.

**Metabolism of PF-00835231 in human liver microsomes: intrinsic clearance and effects of CYP3A inhibitors**. In vitro lability in pooled human liver microsomes was determined as follows. PF-00835231 (1 μM) was incubated with human liver microsomes (2.0 mg/mL; custom mixed sex pool from 50 donors, Xenotech, Lenexa, KS) in a volume of 1 mL potassium phosphate (100 mM; pH 7.5) containing MgCl$_2$ (3.3 mM) and NADPH (1.3 mM) at 37 °C in a shaking water bath. At 0, 5, 10, 20, 30, 45, and 60 min, aliquots (0.1 mL) were removed and added to 0.5 mL acetonitrile containing saquinavir (0.2 μM) as an internal standard to terminate the reaction. Precipitated protein was removed by spinning in a centrifuge ($1700 \times g$; 5 min) and the supernatant was removed in a vacuum centrifuge. The residues were reconstituted in 0.05 mL of 1% formic acid in 20% acetonitrile for analysis by high-performance liquid chromatography-tandem mass spectrometry (HPLC-MS). Intrinsic clearance by substrate depletion was calculated taking into account mg of microsomes per liver weight and grams of liver per Kg body weight using Eq. (3)[54].

$$\mathrm{CL_{int,app}} = \frac{0.693}{\text{in vitro T1/2}} \cdot \frac{\text{ml incubation}}{\text{mg microsomes}} \cdot \frac{45 \text{ mg microsomes}}{\text{gm liver}} \cdot \frac{20 \text{ gm liver}}{\text{Kg.b.w}} \quad (3)$$

Enzyme kinetics were determined by product formation. PF-00835231 (0.25–500 μM) was incubated in pooled human liver microsomes (0.5 mg/mL) in 0.1 M potassium phosphate (100 mM; pH 7.5) containing MgCl$_2$ (3.3 mM) and NADPH (1.3 mM). Incubations were carried out at 37 °C by shaking in a

humidified incubator (85% relative humidity) for 20 min and terminated with the addition of 0.5 mL acetonitrile containing saquinavir (0.1 μM) as an internal standard. Terminated incubation mixtures were processed as described above and analyzed by HPLC-MS. Metabolites M1, M3, and M4 were prepared by biosynthesis using recombinant CYP3A5. PF-00835231 (0.025 mM) was incubated with CYP3A5 (100 pmol/mL) in 40 mL of 100 mM potassium phosphate buffer, (100 mM; pH 7.4) containing MgCl$_2$, (3.3 mM) and NADPH (1.3 mM). Incubations were carried out in a shaking water bath maintained at 37 °C for 1 h. The incubation was terminated with addition of an equal volume of acetonitrile, spun in a centrifuge at $1700 \times g$ for 5 min, and the supernatants subjected to vacuum centrifugation for approximately 1.5 h. To these mixtures was added 0.25 mL each of formic acid and acetonitrile, and water was added to a final volume of 25 mL. This mixture was spun in a centrifuge at $40,000 \times g$ for 30 min. The clarified supernatant was applied to an HPLC column (Polaris C18, 4.6 × 250 mm; 5 μm) at 0.8 mL/min through a Jasco HPLC pump. After application, the column was moved to a Waters Acquity HPLC-UV system coupled with a Thermo LTQ mass spectrometer and CTC Analytics fraction collector. Fractionation of the metabolite mixture was accomplished using a mobile phase comprised of 0.1% formic acid (A) and acetonitrile (B) at a flow rate of 0.8 mL/min. The mobile phase composition commenced at 15% B for 5 min, followed by a gradient to 35% B at 80 min and a second gradient to 95% B at 90 min. Fractions were collected every 20 s. Those containing metabolites M1, M3, and M4 were each combined, evaporated, and analyzed by quantitative NMR spectroscopy[55].

Metabolite M2 was produced in a similar manner but using two separate 20 mL incubations of monkey and hamster liver microsomes (at 2 mg/mL), which were combined for metabolite isolation. Processing of the incubation was as above, however a different HPLC condition was employed. For separation of M2, the mobile phase composition was 2% B for 5 min, 10% B at 6 min followed by a gradient to 40% B at 70 min and a second gradient to 95% B at 90 min. Fractions containing metabolite M2 were pooled, evaporated, and subjected to a second purification step using the same column but an altered mobile phase gradient. In this second fractionation, the initial mobile phase composition was 10% B held for 6 min, followed by two successive gradients to 25% B at 80 min and 95% B at 90 min. Fractions containing M2 were combined, evaporated, and analyzed by quantitative NMR spectroscopy.

Isolated metabolites were dissolved in 0.045 mL of DMSO-d$_6$ "100%" (Cambridge Isotope Laboratories, Andover, MA). Samples were placed in a 1.7 mm NMR tube in a dry argon atmosphere. (For M2, due to the lower amount, a 1.0 mm NMR tube was used with a volume of 0.010 mL DMSO-d$_6$.) $^1$H and $^{13}$C spectra were referenced using residual DMSO-d$_6$ ($^1$H δ = 2.50 ppm relative to TMS, δ = 0.00, $^{13}$C δ = 39.50 ppm relative to TMS, δ = 0.00). NMR spectra were recorded on a Bruker Avance 600 MHz (Bruker BioSpin Corporation, Billerica, MA) controlled by Topspin V3.1 and equipped with a 1.7 mm TCI Cryo probe. 1D spectra were recorded using an approximate sweep width of 8400 Hz and a total recycle time of approximately 7 s. The resulting time-averaged free induction decays were transformed using an exponential line broadening of 1.0 Hz to enhance signal to noise. The 2D data were recorded using the standard pulse sequences provided by Bruker. At minimum a 1 K × 128 data matrix was acquired using a minimum of 2 scans and 16 dummy scans with a spectral width of 10000 Hz in the f2 dimension. The 2D data sets were zero-filled to at least 1k data point. Post-acquisition data processing was performed with MestReNova V9.1.

Experiments to evaluate the role of CYP3A were conducted using a substrate concentration of 4.0 μM, and concentrations of ketoconazole ranging from 1 to 10,000 nM. To evaluate the potential contribution of CYP3A5, the comparative effects of ketoconazole vs cyp3cide (both at 1.0 μM) on the formation of four metabolites of PF-00835231 (at 4 μM) were measured in a custom pool of liver microsomes from four CYP3A5 EM donors, using the same incubation conditions described above. All measurements were made in a minimum of triplicate incubations.

**Metabolic stability of PF-07304814 in human liver S9**. Human liver S9 (HLS9) was purchased from BioIVT (Westbury, NY) as a custom pool of 6 donors pre-pared in the absence of EDTA and PMSF. The experimental conditions used for enzyme kinetics studies in liver S9 were chosen to yield linear reaction velocities as determined from preliminary range finding experiments. Stock solutions of PF-07304814 were prepared in 10/90 acetonitrile/water at 10 times the intended incubation concentration. Incubations were conducted in 100 mM Tris buffer (pH 7.5) containing 5 mM MgCl$_2$ and 0.03 mg/mL S9 protein at 37 °C open to air. The concentrations of PF-07304814 ranged from 0.3 to 300 μM and the final incubation volume was 20 μL. Reactions were initiated by the addition of substrate into enzyme matrix and were terminated after 20 min with 100 μL of acetonitrile containing 1% formic acid and as an internal standard (50 ng/mL indomethacin). To minimize saturation of MS detection, a 10-fold dilution of samples containing the highest substrate concentrations was conducted prior to precipitation. Samples were vortexed, centrifuged, and 75 μL of supernatant was combined with 200 μL of 1% formic acid in water for injection onto LC-MS system for analysis. Incubations for enzyme kinetic determination in liver S9 were conducted in triplicate. Enzyme kinetics were analyzed as described above for HLM using the Michaelis-Menten model with a second unsaturable component. The apparent intrinsic clearance for

PF-07304814 in liver S9 across species was calculated using Eq. (4).

$$CL_{int,app} = \frac{V_{max}}{K_m} \quad (4)$$

**Reversible inhibition of CYP enzymes in human liver microsomes**. CYP selective marker substrates were purchased from various commercial sources: furafylline, ticlopidine, paroxetine, phenacetin, acetaminophen, diclofenac, dextromethorphan and dextrophan were obtained from Sigma Aldrich (St. Louis, MO); S-mephenytoin from Toronto Research Chemicals (North York, ON); verapamil from Sequoia Research Products (Pangbourne, United Kingdom); amodiaquine from Fluka (Bucks, Switzerland); 4'-OH-S-mephenytoin and [2H3]4'-OH-S-mephenytoin from Syncom (Groningen, Netherlands); N-desethylamodiaquine, [2H5]N-desethylamodiaquine, tienilic acid, and [2H3]-dextrophan from Cerilliant (Round Rock, Texas); gemfibrozil glucuronide, 4'-OH diclofenac, and [13C6]4'-OH diclofenac were from Pfizer Inc (Groton, CT). Reversible inhibition was measured in a co-incubation of PF-00835231 with marker substrate (at approximately KM concentration) with HLM (0.01-0.1 mg/ml) and NADPH (1.3 mM) in 100 mM potassium phosphate buffer containing 3.3 mM MgCl$_2$ (pH 7.4) in a final incubation volume of 0.2 mL in a 37 °C dry heat bath open to air. After a 4–20 min incubation time, reactions were quenched with 2-volumes of acetonitrile containing internal standard (2H or 13 C labeled analyte). To qualitatively determine the potential for TDI, all incubation components except marker substrate were preincubated for 30 min, followed by the addition of marker substrate for the specified reaction time. PF-00835231 was tested at concentrations 0.1 to 200 μM in triplicate. Terminated incubation mixtures were centrifuged at 2000 × g for 5 min, resulting supernatants were evaporated to dryness under nitrogen followed by reconstitution in the mobile phase (20% acetonitrile in 1% aqueous formic acid). Samples were analyzed by LC-MS. Analytes were quantified versus a standard curve using GraphPad Prism v8 (San Diego, CA) or Sciex Analyst software. IC$_{50}$ calculations were conducted in GraphPad using Eq. (5).

$$Y = Bottom + \left( \frac{Top - Bottom}{1 + \frac{X^{HillSlope}}{IC_{50}^{HillSlope}}} \right) \quad (5)$$

TDI signal as defined having either at least a 1.5-fold decrease in IC$_{50}$ value after a 30-min preincubation or there was at least a 20% increase in inhibition observed after a 30-min preincubation for any concentration examined. A TDI signal resulted in the determination of $K_I$ and $k_{inact}$. For inhibition studies conducted in cassette format, incubations were conducted as described above with the following modifications: HLM concentration was 0.03 mg/mL, probe substrates bupropion and testosterone were excluded, and the activity reaction time was 6 min.

**Time-dependent inhibition of CYP enzymes in human liver microsomes**. $K_I$ and $k_{inact}$ studies were measured in a co-incubation of PF-00835231 with HLM (0.2-0.3 mg/mL) and NADPH (1.3 mM) in 100 mM potassium phosphate buffer containing 3.3 mM MgCl$_2$ (pH 7.4) in a final incubation volume of 0.2 mL in a 37 °C dry heat bath open to air. After a 1–50-min preincubation time, an aliquot of incubate was diluted 20-fold into prewarmed buffer containing marker substrate (5–10 times above $K_M$ concentration), followed by an activity incubation: 20 μM midazolam (midazolam 1′-hydroxylation) 4 min with 0.2 mg/mL HLM; CYP3A, 386 μM testosterone (testosterone 6β-hydroxylation) 11 min with 0.3 mg/mL HLM. Reactions were quenched with 2-volumes of acetonitrile containing internal standard (²H or ¹³C labeled marker product) and treated as described above for IC$_{50}$ samples. Estimation of the TDI inactivation parameters was performed using Excel and GraphPad software. Percent activity remaining was obtained by normalizing the concentration of marker substrate formed in each sample to the mean solvent control at the first preincubation time point. The natural log (ln) of the percentage remaining activity was plotted against the preincubation time. The slope ($-k_{obs}$, observed rate) of each line was then calculated for the linear portion of the curve and the nonlinear outliers, as determined by GraphPad automatic outlier elimination, were excluded. A statistical test was applied at each inhibitor concentration to evaluate if k$_{obs}$ was significantly different from the solvent control (Eq. (6))[56]:

$$z = \frac{\left| k_{obs[I]} - k_{obs[0\mu M]} \right|}{\sqrt{SE^2_{k_{obs[I]}} + SE^2_{k_{obs[0\mu M]}}}} \quad (6)$$

In this equation, $k_{obs[I]}$, $k_{obs[0 \mu M]}$, and S.E. represent the inactivation rate at each inhibitor concentration, inactivation rate with solvent control, and standard error, respectively. When $p < 0.05$, there was statistically significant or measurable TDI. $K_I$ and $k_{inact}$ were calculated from the nonlinear regression of a 3-parameter Michaelis–Menten equation (Eq. (7)) using GraphPad.

$$k_{obs} = k_{obs[0\mu M]} + \frac{k_{inact} \times [I]}{K_I + [I]} \quad (7)$$

**In vitro transporter inhibition studies using transporter-transfected cells**. HEK293 cells, wild-type and stably transfected with OATP1B1, OATP1B3, OCT1, OCT2, MATE1 and MATE2K, were seeded at a density of 0.5–0.7 × 10⁵ cells/well

on BioCoat™ Poly-D-Lysine 96-well plates (Corning) and grown in Dulbecco's modified Eagle's medium containing 10% FBS, 1% sodium pyruvate, 1% Gluta-MAX™, 1% Gentamicin and 1% nonessential amino acids for 48–72 h at 37 °C, 90% relative humidity, and 5% CO$_2$. For the inhibition studies, HEK293 cells were washed three times with warm transport buffer (Hanks' balanced salt solution with 20 mM 4-(2-Hydroxyethyl) piperazine-1-ethanesulfonic acid, pH 7.4) followed by incubation with test compounds containing probe substrates: 10 or 20 μM [¹⁴C]-metformin (OCT1, OCT2, MATE1, MATE2K), 0.5 μM [³H] para-aminohippuric acid (OAT1), 0.1 μM [³H] estrone-3-sulfate (OAT3) or 0.5 μM rosuvastatin (OATP1B1/1B3). Uptake was terminated by washing cells at least three times with ice-cold transport buffer and then lysing with 0.2 mL of scintillation fluid or 0.225 mL of methanol containing internal standard. For radiolabeled compounds, radioactivity in each sample was quantified by measurement on MicroBeta (Perkin Elmer). For rosuvastatin, cell extracts were dried down under nitrogen and reconstituted in 50:50 (v:v) methanol:water prior to injection onto a LC-MS/MS system. The total cellular protein content was determined by using a BCA Protein Assay Kit (Pierce Technology) following the manufacturer's protocol. Uptake ratio was then derived as a ratio of accumulation in transfected cells to accumulation in wild-type cells. The half-maximal inhibitory concentration (IC$_{50}$) for each transporter was calculated in GraphPad Prism.

(HEK = Human embryonic kidney; OATP = Organic anion transporting polypeptide; OCT = Organic cation transporter; MATE= Multi drug and toxin extrusion protein)

**In vitro transporter inhibition studies using membrane vesicles**. Vesicle assay buffer (10 mM Tris base, 250 mM sucrose, 10 mM magnesium chloride) and stop buffer (assay buffer plus 100 mM sodium chloride) were prepared at pH 7.4. For inhibition assays, HEK293, human BCRP and human MDR1 (P-gp) vesicles were diluted to 50 μg/well in assay buffer and were treated with 5 mM ATP, 0.2 μM rosuvastatin (BCRP), or 0.2 μM N-methyl quinidine (MDR1) and varied concentrations of test compounds. BCRP and MDR1 assays were incubated for 1 and 1.5 min, respectively, at 22 °C while shaking. Reactions were stopped by adding 0.2 mL of ice-cold stop buffer. The entire reaction was quickly removed from the assay plate and filtered on Multiscreen filter plate (Millipore) and then washed four times with ice-cold stop buffer. Intravesicle samples were extracted by treating the vesicles with 0.15 mL of methanol containing internal standard. Vesicles were shaken for 25 min at room temperature. Vesicle extracts were transferred to 96-well polypropylene deep-well plates by centrifugation and dried down under nitrogen. Samples were reconstituted in 50:50 (v:v) methanol:water prior to injection onto a LC-MS/MS system. Percent activity values were exported into GraphPad Prism to estimate IC$_{50}$ values.

(MDR1 = Multi drug resistant AKA P-glycoprotein; BCRP = Breast cancer resistant protein)

**Plasma protein, liver microsome, and liver S9 binding**. Frozen plasma in K$_3$EDTA was purchased from BioIVT and Dulbecco's phosphate buffered saline (DPBS) and HCl were purchased from Sigma. Fraction unbound was determined by equilibrium dialysis using an HTD 96 device (HTDialysis, LLC, Gales Ferry, CT) assembled with 12-14k MWCO membranes. Plasma was thawed and adjusted to pH 7.4 with 1 N HCl prior to use. Dialysis chambers were loaded with 150 μL plasma and 150 μL PBS in the donor and receiver chambers, respectively. Human liver microsomes were diluted to a concentration of 0.8 mg/mL in 100 mM potassium phosphate buffer, donor chambers were loaded with 150 μL diluted HLM and receiver chambers 150 μL potassium phosphate buffer. Human liver S9 was diluted to a concentration of 0.03 mg/mL in 100 mM potassium phosphate buffer, donor chambers were loaded with 150 μL diluted HLS9, and receiver chambers 150 μL potassium phosphate buffer. The dialysis plate was sealed with a gas-permeable membrane and stored in a 37 °C water-jacked incubator maintained at 75% relative humidity and 5% CO$_2$, on a 100 rpm plate shaker. After a 6-h incubation, samples were matrix-matched and quench by protein precipitation, followed by LC-MS analysis. A set of satellite samples was included to measure stability after a 6-h incubation. Incubations were conducted with 4 to 12 replicates. Fu was calculated by dividing the analyte-to-internal standard peak area ratio or analyte concentration in the buffer sample by the signal in the donor sample, corrected for any dilution factors. The HLM binding measured at 1 mg/mL was adjusted to account for the HLM protein concentration (2 mg/mL) using a dilution equation[57]. All incubations had >70% analyte recovery and >70% stability in 6 h.

**Preclinical pharmacokinetic studies**. All activities involving animals were carried out in accordance with federal, state, local and institutional guidelines governing the use of laboratory animals in research in an AAALAC accredited facility and were reviewed and approved by Pfizer's Institutional Animal Care and Use Committee.

Rat PK studies were done at Pfizer (Groton, CT) or BioDuro Pharmaceutical Product Development Inc. (Shanghai, PRC); Jugular vein-cannulated male Wistar-Hannover rats were purchased from Charles River Laboratories, Inc. (Wilmington, MA) or Vital River (Beijing, China) and were typically 7–10 weeks of age at the time of dosing. During the pharmacokinetic studies all animals were housed individually. Access to food and water was provided ad libitum. Compounds were

administered i.v. via the tail vein ($n$ = 2 or 3), dosed as a 1 mg/ml solution using standard compatible excipients (PF-00835231, 2 mL/kg or PF-07304814, 1 mL/kg) for a resulting dose of 2 mg/kg PF-00835231 or 1.17 mg/kg PF-07304814. Serial blood samples were collected via the jugular vein cannula at predetermined timepoints after dosing. Animals were monitored for pain or distress throughout the study, with at least daily monitoring during normal husbandry prior to study start. At the completion of the study, animals were euthanized by overdose of inhaled anesthesia followed by exsanguination. Blood samples were collected into tubes containing K3EDTA and stored on ice until centrifugation to obtain plasma, which was stored frozen at −20 °C or lower. Urine samples were collected at room temperature and stored frozen at 20 °C or lower at the end of each time interval.

Dog PK studies were done at Pfizer (Groton, CT); animal care and in vivo procedures were conducted according to guidelines from the Pfizer Institutional Animal Care and Use Committee. Male Beagle dogs were purchased from Marshall BioResources (North Rose, New York) and were typically 1–5 years of age at the time of dosing. Compounds were administered i.v. via the cephalic vein ($n$ = 2), dosed as a 2 mg/ml solution using standard compatible excipients (PF-00835231, 0.5 mL/kg or PF-07304814, 0.5 mL/kg) for a resulting dose of 1 mg/kg PF-00835231 equivalents or 1.17 mg/kg PF-07304814. Serial blood samples were collected via the jugular vein at predetermined timepoints after dosing. Animals were monitored for pain or distress throughout the study, with at least daily monitoring during normal husbandry prior to study start. Blood samples were collected into tubes containing K3EDTA and stored on ice until centrifugation to obtain plasma, which was stored frozen at −20 °C or lower. Urine samples were collected at room temperature and stored frozen at −20 °C or lower at the end of each time interval.

Non-human primate PK studies were conducted at Pfizer (Groton, CT); All procedures performed on the animals were in accordance with regulations and established guidelines and were reviewed and approved by an Institutional Animal Care and Use Committee through an ethical review process. Male Cynomolgus monkeys were purchased from Covance (Princeton, NJ), Charles River Laboratories, Inc. (Wilmington, MA), or Envigo Global Services (Indianapolis, IN); animals 3–8 years of age were used in PK studies. Compounds were administered i.v. via the saphenous or cephalic vein ($n$ = 2), dosed as a 1 mg/ml solution using standard compatible excipients (PF-00835231, 1 mL/kg or PF-07304814, 0.5 mL/kg) for a resulting dose of 1 mg/kg PF-00835231 or 1.17 mg/kg PF-07304814. PF-00835231 was also administered via oral gavage ($n$=2) as a 1 mg/ml suspension in 0.5% methylcelluose (w/v) in water, at a dose volume of 5 mL/kg, for a final dose of 5 mg/kg. Serial blood samples were collected via the femoral vein at predetermined timepoints after dosing. Animals were monitored for pain or distress throughout the study, with at least daily monitoring during normal husbandry prior to study start. Blood samples were collected into tubes containing K3EDTA and stored on ice until centrifugation to obtain plasma, which was stored frozen at −20 °C or lower. Urine samples were collected at room temperature and stored frozen at −20 °C or lower at the end of each time interval.

Plasma and urine samples were processed using protein precipitation with acetonitrile:methanol containing internal standard propranolol (50 ng/ml) followed by quantitation against a standard curve (0.1–2500 ng/ml) prepared in blank plasma.

Pharmacokinetic parameters were calculated using noncompartmental analysis (Watson v.7.5, Thermo Scientific). The area under the plasma concentration-time curve from t = 0 to infinity (AUCinf) was estimated using the linear trapezoidal rule. Plasma clearance (CLp) was calculated as the i.v. dose divided by AUCinf. The terminal rate constant (kel) was calculated by linear regression of the terminal phase of the log-linear concentration-time curve and the terminal elimination t1/2 was calculated as 0.693 divided by kel. Apparent steady-state distribution volume (Vdss) was determined by clearance multiplied by mean residence time. Percent excreted in urine was calculated as the amount of analyte in the urine divided by the amount dosed. Oral bioavailability was defined as the dose-normalized AUC after oral administration divided by the dose-normalized AUC after i.v. administration. Percent conversion of prodrug to active metabolite was calculated by dividing the dose-normalized PF-00835231 AUCinf after i.v. dosing of prodrug PF-07304814 by the AUCinf after i.v. dosing of PF-00835231 incorporating their respective differences in molecular weight.

**Bioanalytical LC-MS analysis.** Metabolic profiling samples were analyzed by ultrahigh-performance liquid chromatography coupled to UV spectrometry and high-resolution mass spectrometry (UHPLC-UV-HRMS). The system consisted of a Thermo Accela quaternary HPLC pump (Waltham, MA), Thermo Accela diode array UV/VIS detector, and CTC Analytics autoinjector (Zwingen, Switzerland), hyphenated with a Thermo Orbitrap Elite high-resolution mass spectrometer. Samples (0.01 mL) were injected onto a Phenomenex XB-C18 column (2.1 × 100 mM, 2.6 μm) (Torrance, CA) maintained at 45 °C, at a flow rate of 0.4 mL/min. The mobile phase components were A: 0.1% formic acid in water and B: acetonitrile. The initial mobile phase composition of 5% B was held for 0.5 min followed by sequential linear gradients to 70% B at 11 min and 95% B at 13 min. This composition was held for 1 min followed by re-equilibration at initial conditions for 1 min. The eluent was passed through the UV detector (scanning from 200 to 400 nm) before introduction into the mass spectrometer. The mass spectrometer was operated in positive mode with ion source temperatures, potentials, and gas flow settings adjusted to optimize the signal for PF-00835231.

LC-MS analysis of remaining PF-00835231 and PF-07304814 samples was typically performed using tandem liquid chromatography-mass spectrometry (LC-MS/MS) with a Sciex Triple Quad 5500 or 6500 mass spectrometer (Sciex, Framingham, MA), equipped with electrospray sources and Agilent 1290 binary pump (Santa Clara, CA). Aqueous mobile phase (A) was comprised of 1.0% formic acid in water and organic mobile phase (B) consisted of 1.0% formic acid in acetonitrile. Ten μl of sample was injected onto an Acquity UPLC BEH C18 (2.1 × 50 mm, 1.7 μm) (Waters Corporation, Milford, MA) or Halo C18 (2.1 × 30 mm, 2.7 μm) (Advanced Materials, Wilmington, DE) column at room temperature with a flow rate of 0.5 mL/min. The gradient program began with 5% initial mobile phase B for 0.3 min, followed by a linear gradient to 95% B over 2 min, held at 95% B for 0.3 min followed by re-equilibration to initial conditions for 0.6 min. MS was operated in multiple reaction monitoring (MRM) mode, in positive detection mode, with the following mass transitions and collision energies: PF-00835231 473.2/187.0 (CE 16), PF-07304814 553.1/267 (CE 24), propranolol 260.2/116.2 (CE 30), and indomethacin 358.1/139.0 (CE 27).

CYP inhibition samples were analyzed using MRM in positive ion mode. 1A2 reactions were analyzed for acetaminophen 152.1/110.0 and [$^2$H$_7$]-acetaminophen 159.0/115.0 (CE 22) using a Phenomenex Synergi Hydro-RP column (2 × 30 mm, 2.5 μm). 2B6 reactions were analyzed for hydroxybupropion 256.0/139.0 and [$^2$H$_6$]-hydroxybupropion 262.0/139.0 (CE 31) using a Halo C18 column (2.1 × 30 mm, 2.7 μm). 2C8 reactions were analyzed for N-desethylamodiaquine 328.2/283.0 and [$^2$H$_5$]-N-desethylamodiaquine 333.2/283.0 (CE 23) using a Phenomenex Synergi Hydro-RP column (2 × 30 mm, 2.5 μm). 2C9 reactions were analyzed for 4'-hydroxydiclofenac 312.0/266.0 and [$^{13}$C$_6$]- 4'-hydroxydiclofenac 318.0/272.0 (CE 20) using a Halo C18 column (2.1 × 30 mm, 2.7 μm). 2C19 reactions were analyzed for 4'-hydroxymephenytoin 235.2/150.1 and [$^2$H$_3$]- 4'-hydroxymephenytoin 238.1/150.1 (CE 26) using a Halo C18 column (2.1 × 30 mm, 2.7 μm). 2D6 reactions were analyzed for dextrorphan 258.1/201.1 and [$^2$H$_3$]-dextrorphan 261.0/201.1 (CE 31) using a Halo C18 column (2.1 × 30 mm, 2.7 μm). 3A4 reactions were analyzed for 1'-hydroxymidazolam 342.1/324.0 and [$^2$H$_4$]-1'-hydroxymidazolam 346.1/328.0 (CE 30) using a Halo C18 column (2.1 × 30 mm, 2.7 μm) or 6β-hydroxytestosterone 305.2/269.2 and [$^2$H$_3$]- 6β-hydroxytestosterone 308.2/272.7 (CE 25) using a Halo C18 column (2.1 × 50 mm, 2.0 μm).

Analyst software was used to measure peak areas and peak area ratios of analyte to internal standard were calculated. A calibration curve was constructed from the peak area ratios (analyte to internal standard) with a weighted linear (1/x2) regression, from which unknown samples concentrations were calculated.

**Human pharmacokinetic predictions.** *Scaling in vitro human liver microsomal or liver S9 apparent CLint to in vivo CLint.*

$$\text{Microsomal or Liver S9 CLintu, in vivo} = \text{SF}\left(\frac{\text{CLint, app} \cdot \text{PRpLW} \cdot \text{LWpBW}}{\text{fu, inc}}\right) \quad (8)$$

CLint, app = apparent intrinsic clearance, PRpLW = Microsomal or S9 protein per liver weight (45 or 121 mg/g respectively), PRpLW = liver weight per body weight (21 g/kg), fu,inc = HLM or S9 unbound fraction, SF = Pfizer scaling factor (x1.3 for HLM, x1 for S9)

*Scaling in vivo CLint to human plasma clearance.*

$$\text{Human CLh} = \frac{\text{CLintu} \cdot \text{fu, p} \cdot \text{Qh}}{(\text{CLintu} \cdot \text{fu.p}/\text{Rbp} + \text{Qh})} \quad (9)$$

CLintu = scaled in vivo intrinsic clearance, fu,p =plasma fu, Qh = hepatic blood flow (20 mL/min/kg), Rbp = in vitro blood to plasma ratio

*Prediction of PF-00835231 human Vdss.*

$$\text{Human Vdss} = \text{human fu, p} \cdot \left(\frac{\text{animal Vdss}}{\text{animal fu, p}}\right) \quad (10)$$

*Calculation of predicted human half-life.*

$$\text{Human half life} = \left(\frac{0.693 \cdot \text{human Vdss}}{\text{human CL}_h}\right) \quad (11)$$

*Calculation of human dose.*

$$\text{Dose} = \left(\text{CLp(act)} \cdot \left(\frac{\text{Css(act)}}{\text{fm(act)}}\right)\right) / \frac{\text{MW(act)}}{\text{MW(pro)}} \quad (12)$$

Rearranged from published equation factoring in MW differences[58]. act = PF-00835231, pro = PF-07304814, CLp = plasma clearance, Css = steady-state concentration, fm = %conversion from PF-07304814 to PF-00835231, MW = molecular weight (PF-07304814 = 552.5, PF-00835231 = 472.5)

**Rat 24-h continuous intravenous infusion GLP toxicity study.** The study was conducted by Charles River Laboratories for Pfizer Inc, in accordance with the U.S. Department of Health and Human Services, Food and Drug Administration, United States Code of Federal Regulations, Title 21, Part 58: Good Laboratory Practice for Nonclinical Laboratory Studies. Briefly, 8-week old male and female Sprague-Dawley rats (n=15 per sex per group) implanted with femoral catheters exteriorized between the scapulae were assigned to vehicle, or PF-07304814 dose groups (80, 360, and 1000 mg/kg). The vehicle and test article were administered

once on Day 1 as a 24-h continuous IV infusion at a rate of 2.5 mL/kg/h. Main group rats (*n* = 10 per sex per group) were euthanized on Day 2 and recovery/delayed toxicity group (*n*= 5 per sex per group) on Day 14. Study evaluations included detailed clinical observations, body weights, food consumption, functional observation battery, clinical pathology, toxicokinetics, and gross and microscopic pathology.

**Rat 4-day continuous intravenous infusion, exploratory non-GLP toxicity study**. The study was conducted with Pfizer Worldwide R & D, and due to the exploratory nature of this study, compliance with GLP regulations was not required. Male Sprague-Dawley rats (*n*= 3 per group) were administered either vehicle or PF-00835231 (at 24.6 and 246 mg/kg/day) by continuous intravenous infusion via a femoral venous catheter for 4 days, followed by necropsies on Day 5. Study evaluations included clinical observations, body weights, clinical pathology, toxicokinetics, gross pathology and microscopic pathology (control and high dose groups only).

**Bacterial reverse mutation assays**. PF-00835231 and PF-07304814 were tested to evaluate their mutagenic potential by measuring their ability to induce reverse mutations at selected loci of several strains of *Salmonella typhimurium* and at the tryptophan locus of *E. coli* strain WP2 *uvr*A in the presence and absence of an exogenous metabolic activation system using standard protocols[59–61]. These GLP studies were conducted by BioReliance Corporation for Pfizer Inc.

**Effect of PF-00835231 and PF-07304814 on hERG potassium channels**. The in vitro effects of PF-00835231 and PF-07304814 on the hERG (human ether-à-go-go-related gene) channel current (a surrogate for $I_{Kr}$, the rapidly activating delayed rectifier cardiac potassium current[62] were tested using standard protocols. This channel has been selected for evaluation because inhibition of $I_{Kr}$ is the most common cause of cardiac action potential prolongation by non-cardiac drugs[63–65]. Increased action potential duration causes prolongation of the QT interval and has been associated with a dangerous ventricular arrhythmia, *torsade de pointes*[63]. The concentration-response relationship of the effect of PF-00835231 and PF-07304814 on the hERG potassium channel current was evaluated at near-physiological temperature in stably transfected mammalian cells that express the hERG gene using standard protocols. These GLP studies were conducted by Charles River Laboratories for Pfizer Inc.

**Secondary/off-target pharmacology**. Secondary pharmacology studies were conducted by Eurofins Cerep on behalf of Pfizer Inc. The in vitro off-target pharmacology of PF-07304814 and PF-00835231 was individually assessed at 100 μM in a broad target profiling panel which represents targets with known links to potential safety concerns and includes G-protein coupled receptors, ion channels, transporters, and enzymes according to established protocols.

**Hemocompatibility studies**. The effects of PF-07304814 and PF-00835231 on human red blood cell hemolysis and plasma flocculation were evaluated in GLP-compliant studies conducted by Charles River Laboratories on behalf of Pfizer Inc. Briefly, whole blood samples from human volunteers were incubated with a range of concentrations of PF-07304814 and PF-00835231 (in relevant vehicles) and hemolysis was evaluated by determination of whole blood hematocrit, whole blood hemoglobin concentration, plasma hemoglobin concentration, plasma hemolytic index, and visual macroscopic hemolysis assessment. Flocculation was evaluated by determination of the plasma turbidity index and visual flocculation assessment.

**Reporting summary**. Further information on research design is available in the Nature Research Reporting Summary linked to this article.

## Data availability

Data are available in the Source Data file and on www.Figshare.com with the following dois:

**Main manuscript**

**Supplement**

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

## Acknowledgements

The authors would like to thank Sarah Lazzaro, Sumathy Mathialagan, Sangwoo Ryu, Mark West and Emi Yamaguchi (Pfizer) for the transporter inhibition studies. Angela Doran, Chad Limanni, Amanda Plante and Jocelyn Rosado for their in vivo and PK study support (Pfizer). Marcus Ewing (Pfizer) and Gail Johnson (Pfizer) for preformulation studies. Li Hao (Pfizer) for sequence analysis support. Shinji Yamazaki (Pfizer) for PBPK modeling simulations. Daniel Lettiere, Michael Homiski, Michelle Kenyon, Asser Bassyouni, Declan Flynn, William Reagan, Victoria Markiewicz and Stephen Jenkinson for overseeing safety studies, and William Reagan, for expert clinical pathology and pathology support for the toxicology studies. Deli Huang for supplying the HeLa-ACE2 stably transfected cell line. R. Albert for support with the BSL3 facility and procedures at the Icahn School of Medicine at Mount Sinai, New York. Devendra Rai (Pfizer) and Charles Tan (Pfizer) for their help with the editorial process and statistics. Andrew Mesecar acknowledges partial support for this project from federal funds from the National Institute of Allergy and Infectious Diseases, National Institutes of Health, Department of Health and Human Services, under Contract No. HHSN272201700060C. The content is solely the responsibility of the authors and does not necessarily represent the official views of the National Institutes of Health. The study performed in Dr. Jun Wang's laboratory is partially supported by NIH grant (AI147325) and the Young Investigator Award grant from the Arizona Biomedical Research Centre (ADHS18-198859). Scripps work was supported by a grant from the Bill & Melinda Gates Foundation #OPP1107194, and the Scripps Family Impact Fund of the Miramar Charitable Foundation (MCF) MBF is supported by NIH grants R21AI158134, R01 AI148166, R21AI153480, HHSN272201400007C and 75N93019C00051. A DARPA subcontract

#HR0011-20-2-0040, BARDA contract #ASPR-20-01495, and Bill and Melinda Gates Foundation grants # INV-006099 and INV-016638. This research was also partly funded by CRIP (Center for Research for Influenza Pathogenesis), a NIAID supported Center of Excellence for Influenza Research and Surveillance (CEIRS, contract # HHSN272201400008C), by DARPA grant HR0011-19-2-0020, by supplements to NIAID grant U19AI142733 U19AI135972 and DoD grant W81XWH-20-1-0270, and by the generous support of the JPB Foundation, the Open Philanthropy Project (research grant 2020-215611 (5384)), and anonymous donors to AG-S.

## Author contributions

A.D.M., E.L., and B.A. contributed to conceptualization, investigation, analysis, visualization and data curation of inhibition of protease activity against a panel of coronavirus 3CL^pro experiments. A.D.M contributed to funding acquisition, supervision, project administration and resources of these experiments. J.W and C.M contributed to conceptualization, investigation, analysis, visualization and data curation of the thermal shift experiments. JW contributed to funding acquisition, supervision, project administration and resources of these experiments. M.A.B, T.F.G. supervised, designed and carried out antiviral synergy infection experiments. N.B., M.G.K., E.C., A.K.C., and T.F.R. contributed to the design and generation of in vitro antiviral synergy data. J.B., J.H., Y.Z., L.M.A., L.L,S.N., R.O., C.S., R.K., R.Ho., and B.B. contributed to the analysis, interpretation of protease and antiviral data from collaborators and internal data. E.K designed and interpreted the ADME transporter experiments, R.S.O., H.E., R.M.J., E.P.K. contributed to the metabolism, pharmacokinetics and bioanalysis of PF-07304814 and PF-00835231. D.A., J.R.L., S.A.L., M.N.O., and M.T. designed and interpreted formulation experiments and characterization of API properties. M.R.R., M.P., K.O., R.Ho. and D.O. designed and synthesized PF-07304814. R.M.J., B.B., R.S.O., and E.K. contributed to the conceptualization, analysis and calculations for the prediction of human PK and dose estimate for PF-07304814. J.G.S., L.W.U., R.M.J. contributed to the design, supervision and interpretation of in vitro and in vivo safety study data. C.A., A.A., M.I.R. contributed to the scientific discussion, experimental design, data interpretation in addition to manuscript review and editing. N.S. contributed to the analysis, interpretation of protease and antiviral data. S.W.M., A.A., and M.B.F. designed and interpreted and H.H., J.L., M.M., R.E.Ha., S.W. executed the SARS-CoV in vivo animal efficacy studies. R.R., K.M.W., and A.G.S. developed the SARS-CoV-2 murine model; R.R., K.M.W., S.W.M., A.A., and A.G.S. designed the SARS-CoV-2 animal study that was executed by K.R.R. and K.M.W. All authors contributed to writing or reviewing drafts of the manuscript.

## Competing interests

A.D.M. and affiliates (B.J.A., E.K.L.) have a sponsored program contract with Pfizer to test compounds for inhibition of coronavirus proteases. J.W. and affiliate (C.M.) have a sponsored research agreement with Pfizer to test compounds for inhibition of coronavirus proteases. The Frieman Laboratory (M.B.F., H.H., R.E.Ha., J.L., M.E.M., S.W.) was funded by Pfizer for the work in this manuscript. The García-Sastre Laboratory (AG-S, KMW, RR) has received research support from Pfizer, Senhwa Biosciences, Accurius, Avimex, 7Hills Pharma, Pharmamar, Blade therapeutics, Dynavax, Kenall Manufacturing, ImmunityBio and Nanocomposix; and AG-S has consulting agreements for the following companies involving cash and/or stock: Vivaldi Biosciences, Contrafect, 7Hills Pharma, Avimex, Valneva, Accurius, and Esperovax. AG-S is inventor in patent applications on SARS-CoV-2 antivirals owned by Icahn School of Medicine at Mount Sinai. B.B., R.M.J., D.A., L.A., J.B., H.E., J.H., R.Ho., E.P.K., R.K., E.K., L.L., J.R.L., S.A.L., S.W.M., S.N., R.S.O., M.N.O., R.O., K.O., D.O., M.P., M.R.R., M.I.R., J.G.S., N.S., C.S., M.T., L.W.U., Y.Z., A.A., and C.A., at the time of their contributions were employees and may have stock in Pfizer. A.K.C., M.A.B., N.B., E.C., M.G.K., T.F.R. have no competing interest to report.
