## [Peer Review File · Nature Communications]

Preclinical Characterization of an Intravenous Coronavirus 3CL Protease Inhibitor for the Potential Treatment of COVID19Reviewers' Comments:

Reviewer #1:

Remarks to the Author:

This study designed and developed a novel phosphate prodrug PF-07304814 for SARS (SARS-CoV-2), which can be metabolized (dephosphorylated) within human body to produce a small molecule inhibitor PF-00835231 for both SARS-CoV-1 and SARS-CoV-2 based largely on a previous study authored by their colleagues. With multiple experiments in vivo and in vitro, this article verifies that PF-07304814 is potential to treat the new coronavirus. The authors conducted the ADME/T prediction of this phosphorylated prodrug and related physical property optimization experiments in a variety of animals, indicating the safety of the prodrug PF-07304814 and predicting the feasibility of clinical treatment. The paper also pointed out that the drug must be metabolized in the body to have an antiviral effect. After metabolism, the metabolite PF-00835231 exhibits extensive and effective anti-coronavirus activity but has a low effect on human proteases and HIV protease through directly binding with 3CLpro. PF-00835231 also exhibited synergy effect on the current new coronavirus with drug Remdesivir. This study is innovative and has a significant effect on the development of clinical new coronavirus treatment drugs. However, I have the following issues which the authors should address clearly in their revision:

Major Concerns:

1. The synergistic, additive and antagonistic effects of the prodrug and Remdesivir were detected in the serum of clinical patients. But only two patient serums were tested and the results respectively showed a synergistic and an additive effect of the combination drugs on the antiviral activity against SARS-CoV-2. Though the tests fortunately did not provide an antagonistic effect, the sample amount is too small and it is strongly suggested that the authors need to test many more samples (at least $3 \times 3 = 9$ patient serums).
2. In my opinion, it is not quite reasonable for the authors to use SARS-CoV-1 infected mouse model (SARSCoV-MA15) in testing the in vivo activity of PF-00835231 since this study is the discovery of the novel drug for the potential treatment of COVID-19. Why the authors did not use SARS-Cov-2 infected mouse models, such as that published in the literature (A mouse-adapted model of SARS-CoV-2 to test COVID-19 countermeasures, Nature, 2020, 586: 560–566, 2020)?
3. In several places, the authors stated the affirmative results but failed to provide either the experimental data or literature citations. For examples, in page 7, the authors wrote that "The P-gp inhibitor alone had no antiviral or cytotoxic activity at these concentrations and did not cause cytotoxicity in the presence the protease inhibitor." In page 9, the authors described that "The observed additivity/synergy was not due to cytotoxicity, as there was no noticeable cytotoxicity in virus infected host cells for all the combinations tested."

Minor concerns:

1. Some figures are obviously blurry and unclear, such as Fig2, Fig3B and Fig6. The authors need to improve the figure qualities in their revision.
2. There are some grammar mistakes or inappropriate format throughout the whole manuscript. A simple example is that a space needs to be always given between the value and the unit. The authors may ask Professional Editing Service such as Wordvice or AJE for a revision.

Reviewer #2:

Remarks to the Author:

In the present work, Boras et al. report the 3CLpro inhibitor, PF-00835231, its novel phosphate prodrug, PF-07304814, and present broad-spectrum inhibitory activity across coronaviral 3CL proteases. The biological, chemical, and structural features of PF-00835231 have mostly been

described in previous reports and the main body of the present manuscript focuses on its pharmacological features (ADME). The synthesis of PF-00835231's phosphate prodrug (PF-07304814) is also described.

Major points

1. The PK data in Table S8 show that the terminal $t_{1/2}$ values for PF-00835231 are as short as 1.4 h in rat, 1.5 h in dog, and 1.2 h in monkey. Despite of these short terminal $t_{1/2}$ values, the authors suggest a continuous IV infusion as the optimal dosing route. Considering that the PF-00835231 doses at 100 mg/kg and 300 mg/kg apparently effected in MA15-infected mice (Figure 4), extremely high doses of PF-00835231 appear to be needed for treatment. Moreover, if one carefully looks at the PK data in Figure 6A, plasma half-life values of PF-00835231 appear to be as short as 10-20 minutes or such in both "prodrug-active" and "active-active" examinations. The authors project the PK of continuous IV infusion of PF-00835231 in Figure 6C. However, considering that PF-00835231 is quickly decomposed/degraded, the authors should rather determine actual PK data with continuous IV infusion of PF-00835231 in any animals available to them.

Minor points

1. There is a critical typo on third line in Abstract: PF-00835321 should be PF-00835231.
2. The numbers of the infected and death toll of COVID-19 are outdated back to December 2020. These data should be updated.
3. On page 7, the authors say that viral proteins were detected in the assay using two convalescent human polyclonal sera; however, the nature of the assay is not mentioned. Did the authors use ELISA or immunohistochemistry and how?
4. Although the authors show that histological changes were seen in HE-stained lung tissues, they do not show the reduction of virally-infected cells. The authors should attempt to examine if the lung tissues they used for HE staining contain SARS-CoV-2-infected cells with or without PF-00835231.
5. Typographic errors are seen in various spots on page 44. "mgCl2" should be "magnesium chloride" or "MgCl2". "IS" should be spelled out as "internal standards".

Reviewer #3:

Remarks to the Author:

The manuscript from Jones et al. reports on the discovery of a protease inhibitor, and its prodrug, as clinical candidates in the treatment of COVID-19.

The data disclosed is very interesting and of high scientific value for researchers in the field, and beyond. The described compound is currently in clinical evaluation (Phase I), and it represents the first in this class to reach this stage. Based on the PK related data presented, and the need of an IV infusion, it is fair to say that, even if approved, the compound does not represent an ideal drug. However, the significance of this work, in this moment in time, cannot be underestimated. Despite the vaccine successes, an effective antiviral therapy (that would likely be a combination therapy), will be essential in the future covid-19 management, and potentially to other related coronaviruses infections that might arise in the future. The compounds reported are an extremely important step in that direction.

The work as presented, has few minor limitations. In particular: the use of the animal model that is not infected with SARS-CoV-2. However, considering how the model is used, and, importantly, the high similarity of the viral target, between SARS-CoV and SARS-CoV-2, the data gives enough confidence on the potential therapeutic value on the reported compounds against COVID-19.

A second limitation, which I believe it should be addressed, is the lack of data on potential SARS-CoV-2 resistance against the active compound. It would be extremely interesting and useful to understand if specific mutation arise, how quickly they develop, and the fitness of the resistant viruses.

Finally, the manuscript is well written and presented (there is a misplaced parenthesis on page 7, line 4. I believe it should not be at the end one the sentence, but after "cess", maybe).

Overall, this is an important paper that should be considered for publication.

Response to reviewers' comments:

Reviewer 1:

1. The synergistic, additive and antagonistic effects of the prodrug and remdesivir were detected in the serum of clinical patients. But only two patient serums were tested and the results respectively showed a synergistic and an additive effect of the combination drugs on the antiviral activity against SARS-CoV-2. Though the tests fortunately did not provide an antagonistic effect, the sample amount is too small and it is strongly suggested that the authors need to test many more samples (at least $3 \times 3 = 9$ patient serums).

Response: Thank you for this comment. The patient serum was not used to evaluate the effects of the prodrug or remdesivir. The patient serum was used strictly as a detection reagent in the assay. That is, the cells were treated with compounds of interest, infected with SARS-CoV-2, fixed and then stained with patient serum to detect infected cells. The two patient sera were chosen based on the brightness and specificity of the staining of infected cells, and therefore for their utility as detection reagents. We used the two different sera to confirm that we did not observe serum-specific detection artefacts in our assay. Additional clarity was added to the methods section (page 43) to reflect this clarification. This method was recently published in (1) This reference was added to the paper to provide additional context.

(1) Garcia, Gustavo et al. Antiviral drug screen identifies DNA-damage response inhibitor as potent blockers of SARS-CoV-2 replication. Cell Reports April 2021;35:1.

2. In my opinion, it is not quite reasonable for the authors to use SARS-CoV-1 infected mouse model (SARSCoV-MA15) in testing the in vivo activity of PF-00835231 since this study is the discovery of the novel drug for the potential treatment of COVID-19. Why the authors did not use SARS-Cov-2 infected mouse models, such as that published in the literature (A mouse-adapted model of SARS-CoV-2 to test COVID-19 countermeasures, Nature, 2020, 586: 560–566, 2020)?

Response: Unfortunately, the ability to evaluate in vivo efficacy in the SARS-CoV-2 model was not available at the time the SARS-CoV mouse efficacy studies were completed. Since the active binding site of SARS-CoV and SARS-CoV-2 are identical we believe this was a valid approach to evaluate the efficacy in this model as a surrogate for SARS-CoV-2 efficacy.

In response to your concern we have been able to generate SARS-CoV-2 mouse efficacy data and is included in the manuscript in Figure 4. The new data, derived from an Ad5-hACE2 SARS-CoV-2 mouse model, shows that PF-00835231 at 100 mg/kg (S.C., BID) reduced viral load by $\approx 1.5 \log_{10}$, which is consistent with reductions observed in the SARS-CoV mouse model.

3. In several places, the authors stated the affirmative results but failed to provide either the experimental data or literature citations. For examples, in page 7, the authors wrote that “The P-gp inhibitor alone had no antiviral or cytotoxic activity at these concentrations and did not cause cytotoxicity in the presence the protease inhibitor.”

Response: Thank you for highlighting this omission. The antiviral and cytotoxicity activity of PF-00835231 was evaluated in a VeroE6 CPE assay in the presence or absence of a P-gp inhibitor, CP-100356. The activity of CP-100356 by itself was assessed with an EC₅₀ of 23.4 μM and CC₅₀ of 29.6 μM. These values are an order of magnitude higher than those tested in combination with PF-00835231. In the presence of 2 μM P-gp inhibitor, the potency of PF-00835231 increased 78-fold without cytotoxicity observed up to 50 μM. In summary, at the doses tested, CP-100356 is not contributing to either antiviral or cytotoxicity activity of PF-00835231.

This information was added to the methods (page 42-43), the table was added as Table S3, and the table was referenced in the text.

Table S3. Summary of the In Vitro Antiviral Activity, Cytotoxicity, and Therapeutic Index for PF-00835231 with and without P-gp Inhibitor

Virus Strain ^a	Host Cell	Compound	P-gp Inhibitor ^b (uM)	EC ₅₀ (uM)	CC ₅₀ (uM)	TI ^c
SARS-CoV2	Vero	CP-100356 ^d	-	23.4	29.6	1.27
SARS-CoV2	Vero	PF-00835231 ^d	0	35.9	>50	1.39
SARS-CoV2	Vero	PF-00835231 ^e	0.5	2.36	>50	21.2
SARS-CoV2	Vero	PF-00835231 ^e	1	0.95	>50	52.6
SARS-CoV2	Vero	PF-00835231 ^d	2	0.46	>50	109

a. Data generated at Southern Research Institute (SRI) – 2020. SARS-CoV-2 Washington strain.

b. P-gP Inhibitor is CP-100356.

c. The TI was calculated by dividing the individual CC₅₀ by the EC₅₀ values and then calculating TI mean.

d. Values are averages based on an n=2.

e. Values are the geometric means based on n=4.

4. In page 9, the authors described that “The observed additivity/synergy was not due to cytotoxicity, as there was no noticeable cytotoxicity in virus infected host cells for all the combinations tested.”

Response: Thank you for this comment. We used host cell counts (number of nuclei per well) as a proxy readout for cytotoxicity. We now specify in the main text that cytotoxicity was evaluated “based on counts of host cell nuclei per well in the infection assay, data not shown” on page 8.

5. Some figures are obviously blurry and unclear, such as Fig2, Fig3B and Fig6. The authors need to improve the figure qualities in their revision.

Response: High resolution figures were not submitted in the previous version based on the guidelines but have been added as requested to the revised manuscript. Further refinement of the Figures can be made as needed.

6. There are some grammar mistakes or inappropriate format throughout the whole manuscript. A simple example is that a space needs to be always given between the value and the unit. The authors may ask Professional Editing Service such as Wordvice or AJE for a revision.

Response: Thank you for your comment. We have added spaces between values and their units (did not highlight everywhere where we did this) and have corrected other typos throughout the manuscript where we found them.

Reviewer 2:

- 1) The PK data in Table S8 show that the terminal $\frac{1}{2}$ values for PF-00835231 are as short as 1.4 h in rat, 1.5 h in dog, and 1.2 h in monkey. Despite of these short terminal $\frac{1}{2}$ values, the authors suggest a continuous IV infusion as the optimal dosing route.

Response: We agree that PF-00823521 does exhibit a short half life and is the reason why a continuous IV infusion is proposed. The predicted half life for human if delivered via an IV bolus is short (~2h). Therefore, in human a constant IV infusion will be needed to achieve and maintain the desired efficacious concentration. Continuously infusing the drug will enable us to overcome the short half-life and provide a flat steady state concentration of PF-00835231 as illustrated in Figure 6c. The use of a continuous IV infusion enables the delivery of the molecule at the same rate it is cleared from the body providing the ability to maintain a constant target concentration.

- 2) Considering that the PF-00835231 doses at 100 mg/kg and 300 mg/kg apparently effected in MA15-infected mice (Figure 4), extremely high doses of PF-00835231 appear to be needed for treatment.

Response: The reason the projected human dose is much lower than used in mouse in vivo efficacy studies is due to the difference in the rate of metabolism and route of administration between mouse (subcutaneous) and human (constant IV infusion). Unfortunately, constant IV infusions are unable to be delivered in the in vivo efficacy mouse model and higher SC doses are required to maintain a C_{min} concentration. For small molecules higher doses are often required in animals to achieve equivalent exposures in human, see following reference for additional information on animal vs human dose considerations (Nair, 2016).

Nair AB, Jacob S. A simple practice guide for dose conversion between animals and human. J Basic Clin Pharma 2016;7:27-31

- 3) Moreover, if one carefully looks at the PK data in Figure 6A, plasma half-life values of PF-00835231 appear to be as short as 10-20 minutes or such in both “prodrug-active” and “active-active” examinations.

Response: Thank you for the query regarding these graphs. When looking at the PK curves in Figure 6a there may be some confusion between the distribution phase and the elimination half life of PF-00835231. PF-00835231 exhibits a biphasic PK profile exhibiting an initial distribution phase followed by an elimination phase. The distribution phase does have a steeper slope with an even shorter half-life (10-20 mins) which represents the drug distributing into tissues, this is not a concern for the molecule and does not represent the half life. The second phase is the elimination phase and is from where the terminal half life is derived (presented in Table S9). In human when delivered as a constant infusion a flat steady state concentration will be maintained which will be achieved when the input rate (via IV infusion) and the output rate (clearance of the molecule from the body) are equivalent.

- 4) The authors project the PK of continuous IV infusion of PF-00835231 in Figure 6C. However, considering that PF-00835231 is quickly decomposed/degraded, the authors should rather determine actual PK data with continuous IV infusion of PF-00835231 in any animals available to them.

Response: PF-00835231 is metabolized by cytochrome P450 enzymes in the liver and the rates of metabolism are not dependent on the administration route or length of infusion. An IV bolus is an appropriate method and a standard approach for characterizing the PK parameters of clearance, volume of distribution and elimination half life for PF-00835231. The PK parameters for PF-00835231 derived following an IV bolus or IV infusion would be expected to be equivalent. For simplicity and to minimize animal restraint IV bolus is the typical administration used to characterize PK parameters in preclinical species. Therefore, completing additional PK characterization with IV infusions in animals were deemed unnecessary and were not completed as part of the characterization of PF-00835231. The animal IV bolus data has not been referred to as supporting the need for an IV infusion in human within the manuscript and therefore additional IV infusion data would not add value to the approach taken. The animal in vivo PK data is primarily presented to support the high conversion from the prodrug PF-07304814 to its active PF-00835231 in vivo and would be expected to be the same following a bolus or infusion delivery.

The prediction of human clearance for PF-00835231 (Figure 6b) does not incorporate the animal PK data and was made using in vitro human liver microsomes as outlined in the supplemental using the approach published by Obach, 2011.

The concept of an IV bolus providing equivalent PK parameters to an IV infusion is an established perspective based on standard PK principles. As a clinical example I refer to midazolam studies where the PK parameters were derived in healthy volunteers as an IV bolus (Heizmann et al, 1983) and as a continuous IV infusion (Malacrida et al, 1991) with both providing equivalent Clearance, Vdss and elimination half lives for midazolam.

Heizmann P, Eckert M, Ziegler W.H, Pharmacokinetics and bioavailability of midazolam in man, Br.J. clin. Pharmac. (1983), 16, 43S-49S

Malacrida R, Fritz M.E, Suter P.M, Crevoisier C, Pharmacokinetics of midazolam administered by continuous infusion to intensive care patients, Critical Care Medicine, 31 Jul 1992, 20(8):1123-1126

To help clarify the need for need for a constant IV infusion an additional sentence was added to the following paragraph in the manuscript (red text).

Page 12-13 of the manuscript:

Due to the rapid blood perfusion through the lungs and the continuous steady state intravenous infusion regimen, the unbound plasma and unbound lung concentrations are assumed to be in equilibrium and, therefore, the unbound plasma concentration provides a reasonable surrogate for the concentration at the main site of action of the disease. Based on the human PK predictions, the minimally efficacious dose of PF-00835231 necessary to achieve this exposure is ~320 mg/day administered as and intravenous continuous infusion. **Due to the projected short elimination half life of PF-00835231 following an IV bolus administration a continuous IV infusion is needed to maintain efficacious concentrations.** The required duration of dosing for efficacy remains uncertain and will need to be evaluated in humans.

- 5) There is a critical typo on third line in Abstract: PF-00835321 should be PF-00835231.

Response: We have corrected this typo.

- 6) The numbers of the infected and death toll of COVID-19 are outdated back to December 2020. These data should be updated.

Response: Thank you, this has been updated to provide numbers that are not date specific.

- 7) On page 7, the authors say that viral proteins were detected in the assay using two convalescent human polyclonal sera; however, the nature of the assay is not mentioned. Did the authors use ELISA or immunohistochemistry and how?

Response: Thank you for this comment. We have now clarified the nature of the assay (immunofluorescent detection of SARS-CoV-2 infected cells using high-content imaging) in the main text (page 7). The assay is also described in detail in the materials and methods section, but as multiple infection assays are referred to in the manuscript, we understand the need for exactness within the main text, which we added. Further, this method was recently published in (Garcia 2021). This reference was added to the paper to provide additional context.

Garcia, Gustavo et al. Antiviral drug screen identifies DNA-damage response inhibitor as potent blockers of SARS-CoV-2 replication. Cell Reports April 2021;35:1.

- 8) Although the authors show that histological changes were seen in HE-stained lung tissues, they do not show the reduction of virally-infected cells. The authors should attempt to examine if the

lung tissues they used for HE staining contain SARS-CoV-2-infected cells with or without PF-00835231.

Response: We have now used IHC to detect SARS-CoV in the lung tissues from infected and untreated, uninfected, and infected and treated mice. The new panels added to Fig 4D indicates that SARS-CoV-infected cells are eliminated from the lungs of animals treated with 300 mg/kg PF-00835231.

- 9) Typographic errors are seen in various spots on page 44. “mgCl₂” should be “magnesium chloride” or “MgCl₂”. “IS” should be spelled out as “internal standards”.

Response: Thank you for this catch. We have changed the MgCl₂ and wrote out internal standards on page 44.

Reviewer 3:

- 1) In particular: the use of the animal model that is not infected with SARS-CoV-2. However, considering how the model is used, and, importantly, the high similarity of the viral target, between SARS-CoV and SARS-CoV-2, the data gives enough confidence on the potential therapeutic value on the reported compounds against COVID-19.

Response: Thank you for this comment. We agree with this assessment. To confirm, we included SARS-CoV-2 data on page 9-10 and in Figure 4, which confirmed a consistent response in the SARS-CoV-2 model with what was observed in the SARS-CoV model.

- 2) A second limitation, which I believe it should be addressed, is the lack of data on potential SARS-CoV-2 resistance against the active compound. It would be extremely interesting and useful to understand if specific mutation arise, how quickly they develop, and the fitness of the resistant viruses.

Response: We see the enormous value of this work, however it is beyond the scope of this paper. We added a comment to the end about the value of this work.

- 3) Finally, the manuscript is well written and presented (there is a misplaced parenthesis on page 7, line 4. I believe it should not be at the end one the sentence, but after “cess”, maybe).

Response: Thank you for this catch. This has been corrected on page 7 line 4.

Reviewers' Comments:

Reviewer #1:

Remarks to the Author:

I have no further concerns and the manuscript can be published in its current form.

Reviewer #2:

Remarks to the Author:

The authors of the MS entitled "Discovery of a Novel Inhibitor of Coronavirus 3CL Protease for the Potential Treatment of COVID-19" have well responded to this Reviewer's queries re the PK issues and the route of administration mentioned in the original version of the manuscript. The authors also appropriately responded to this Reviewer's minor queries/suggestions. This reviewer has no further criticisms or queries in the revised version of the manuscript.